# Combined transcriptome and proteome profiling reveals specific molecular brain signatures for sex, maturation and circalunar clock phase

Sven Schenk[1,2†]*, Stephanie C Bannister[1,2†‡]*, Fritz J Sedlazeck[3§], Dorothea Anrather[1,4], Bui Quang Minh[3#¶], Andrea Bileck[2,5]**, Markus Hartl[1,4], Arndt von Haeseler[2,3,6], Christopher Gerner[2,5], Florian Raible[1,2]*, Kristin Tessmar-Raible[2]*

[1]Max F Perutz Laboratories, University of Vienna, Vienna BioCenter, Vienna, Austria; [2]Research Platform 'Rhythms of Life', University of Vienna, Vienna BioCenter, Vienna, Austria; [3]Center of Integrative Bioinformatics Vienna, Max F Perutz Laboratories, University of Vienna, Medical University of Vienna, Vienna BioCenter, Vienna, Austria; [4]Mass Spectrometry Facility, Max F Perutz Laboratories, Vienna, Austria; [5]Department of Analytical Chemistry, University of Vienna, Vienna, Austria; [6]Bioinformatics and Computational Biology, Faculty of Computer Science, University of Vienna, Vienna, Austria

*For correspondence:
sven.schenk@univie.ac.at (SS);
stephanie.bannister@univie.ac.at (SCB);
florian.raible@mfpl.ac.at (FR);
kristin.tessmar@mfpl.ac.at (KT-R)

†These authors contributed equally to this work

Present address: ‡Lexogen GmbH, Campus Vienna Biocenter, Vienna, Austria; §Human Genome Sequencing Center, Baylor College of Medicine, Houston, United States; #Ecology and Evolution, Research School of Biology, Australian National University, Canberra, Australia; ¶Research School of Computer Science, Australian National University, Canberra, Australia; **Department of Biomedical Research, University Hospital Bern, Bern, Switzerland

**Abstract** Many marine animals, ranging from corals to fishes, synchronise reproduction to lunar cycles. In the annelid *Platynereis dumerilii,* this timing is orchestrated by an endogenous monthly (circalunar) clock entrained by moonlight. Whereas daily (circadian) clocks cause extensive transcriptomic and proteomic changes, the quality and quantity of regulations by circalunar clocks have remained largely elusive. By establishing a combined transcriptomic and proteomic profiling approach, we provide first systematic insight into the molecular changes in *Platynereis* heads between circalunar phases, and across sexual differentiation and maturation. Whereas maturation elicits large transcriptomic and proteomic changes, the circalunar clock exhibits only minor transcriptomic, but strong proteomic regulation. Our study provides a versatile extraction technique and comprehensive resources. It corroborates that circadian and circalunar clock effects are likely distinct and identifies key molecular brain signatures for reproduction, sex and circalunar clock phase. Examples include prepro-whitnin/proctolin and ependymin-related proteins as circalunar clock targets.
DOI: https://doi.org/10.7554/eLife.41556.001

## Introduction

Appropriate timing is essential for reproductive success, especially in organisms that reproduce via external fertilisation. In order to maximise the chances of successful mating encounters, marine animals often rely on mass spawning events that require synchronisation of behaviour and gonadal maturation, both within individuals and on a population level (*Fischer, 1984*; *Guest, 2008*; *Hoeger et al., 1999*; *Tessmar-Raible et al., 2011*).

Many species achieve this synchronisation through the interplay of multiple timing systems that operate on different time scales, ranging from months or days to hours or even minutes (reviewed in *Liedvogel et al., 2011*; *Tessmar-Raible et al., 2011*). One timing cue used for synchronised

**eLife digest** Like many other sea creatures, the worm *Platynereis dumerilii* reproduces by dispersing eggs and sperm in the water. For these animals, timing is everything: if they fail coordinate their release, the precious reproductive cells will drift in the vastness of the ocean without ever meeting their male or female counterparts.

Internal clocks are a set of mechanisms that allow organisms to tune their internal processes to their environment. For example, the circadian clock helps many creatures to adapt to the cycle of day and night. This involves switching genes on and off according to the time of day. When a gene is activated, its information is copied into a molecule of RNA, which is then read to create proteins that will go on performing specific roles. To produce their eggs and sperm at the right time, *P. dumerilii* worms rely on a poorly understood internal clock which is synchronized by the moon cycle.

To investigate this 'inner calendar', Schenk, Bannister et al. developed a new technique that allows them to extract both RNA and proteins from the miniscule heads of the worms. The results showed that the internal clock synchronized by the lunar phases influenced the levels of many more proteins than RNA molecules. In comparison, other life events such as the worms becoming sexually mature, had a more similar impact on both protein and RNA regulation. This might suggest that the inner calendar that coordinates the worms with the moon cycle could work by changing protein, rather than RNA levels. The analysis also highlighted several molecular actors that may be essential for the worm's inner clock to work properly. In the future, the new technique will help to dissect more finely how *P. dumerilii* and many other marine creatures stay synchronized with the moon, and spawn at the right time.

DOI: https://doi.org/10.7554/eLife.41556.002

reproduction in organisms ranging from brown algae and corals to bristle worms, echinoderms and vertebrates, is provided by the changing phases of the moon (*Brady et al., 2016*; *Coppard and Campbell, 2005*; *Grant et al., 2009*; *Kaniewska et al., 2015*; *Kennedy and Pearse, 1975*; *Oldach et al., 2017*; *Rahman et al., 2004*; *Raible et al., 2017*; *Saavedra and Pousão-Ferreira, 2006*; *Saigusa, 1988*; *Skov et al., 2005*). Depending on the species, the predominant physical cue interpreted by organisms is either gravitation or illumination. A study in reefs at two distinct lunar time points (full moon vs. new moon) has revealed a direct impact of moonlight illumination on the transcriptome of *Acropora* corals, but also found that light-induced transcriptome changes are most dramatic in mature animals, when light orchestrates the actual spawning event (*Kaniewska et al., 2015*). While pointing at the transcriptome as a readout for organismal changes, this study therefore also illustrates that in natural samples, the molecular changes relating to illumination can be tightly interwoven with internal changes (maturation and spawning), and might in addition mask endogenous monthly oscillations as they can be produced by endogenous timing systems.

Laboratory research on animals such as the marine annelid *Platynereis dumerilii* or the marine midge *Clunio marinus* has indeed revealed evidence for such internal timers, showing that nocturnal light (in nature provided by the moon) does not only impact directly on transcription (*Zantke et al., 2015*), but is both necessary and sufficient to entrain endogenous monthly (circalunar) oscillators (*Kaiser et al., 2016*; *Zantke et al., 2013*), which in turn orchestrate reproduction at distinct times of the lunar month (*Figure 1a*).

In *Platynereis dumerilli*, sexually mature animals reproduce by engaging in stereotypical swimming behaviours classically referred to as the 'nuptial dance', the exact onset of which is signalled by sex pheromones and results in the coordinated release of gametes by male and female worms (*Beckmann et al., 1995*; *Zeeck et al., 1994*; *Zeeck et al., 1998*). Sexual dimorphisms exist in the production and response of mature worms to spawning pheromones, as well as on the level of gonadal and parapodial morphology. These differences are not obvious during the early and juvenile developmental stages, but develop within a few weeks to months prior to spawning (*Figure 1a,b*). The sex ratio of the worms remains constant within laboratory cultures, even under different temperature, feeding or light conditions and different population densities, consistent with genetic mechanisms underlying the sexual dimorphisms (*Beckmann et al., 1995*; *Fischer and Dorresteijn, 2004*; *Schulz et al., 1989*; *Zeeck et al., 1994*; *Zeeck et al., 1998*).

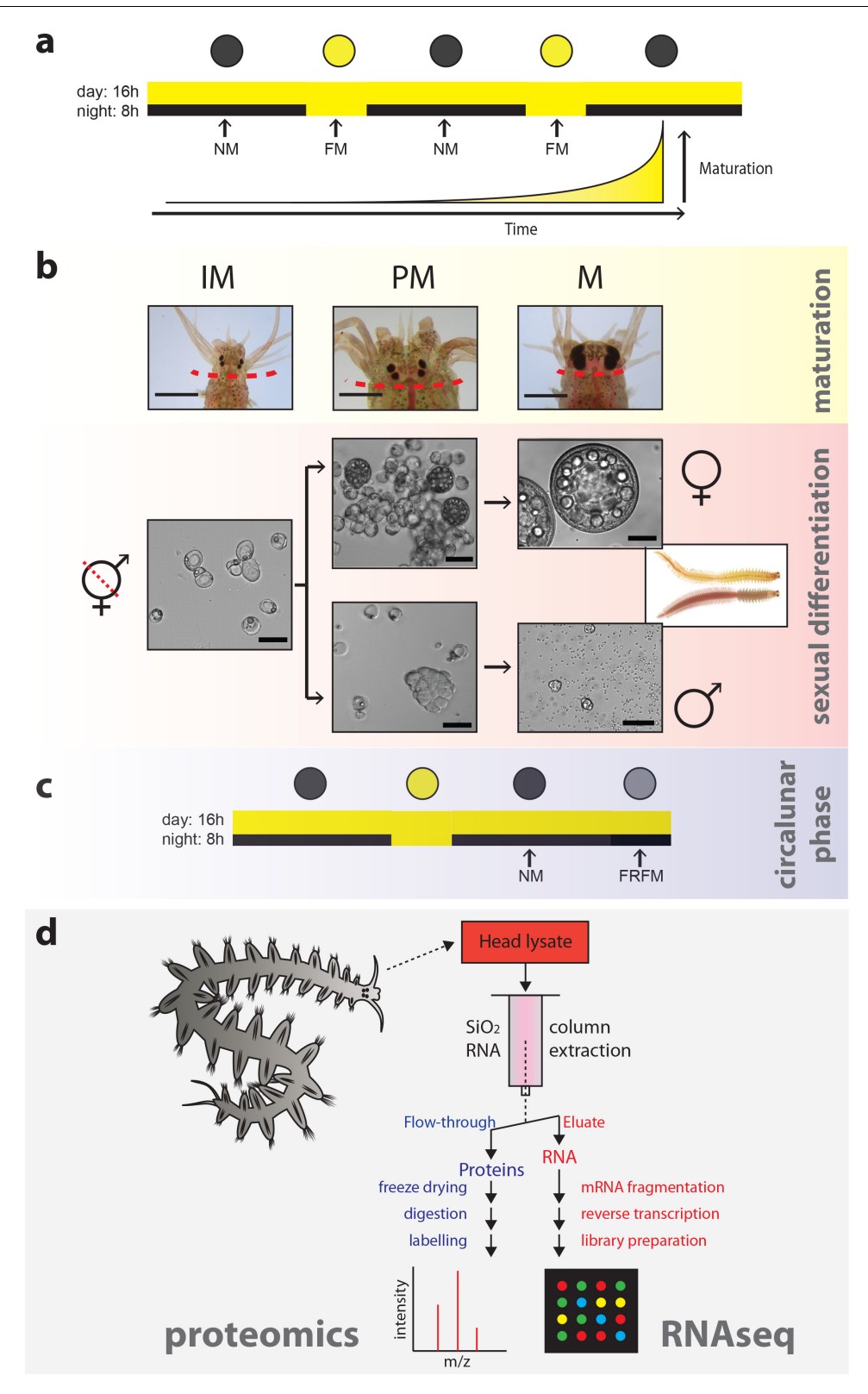

**Figure 1.** Strategy to capture the impact of maturation, sex and circalunar phase on the head transcriptome and proteome of *Platynereis dumerilii*. (a) Light regime of the *Platynereis* culture and its correlation with maturation; daily and nocturnal light conditions are schematised. Eight consecutive nights with nocturnal light per month provide a simplified 'full moon' (FM) stimulus, alternating with dark nights ('new moon'/NM) (intensity and spectra see *Figure 1—figure supplement 2a–d*). (b) Representative images of the different animal stages relevant for the study: IM - immature; PM -

*Figure 1 continued on next page*

*Figure 1 continued*

premature; M - mature. Top row: overview of heads at the respective stage; dotted line: cut site for head sampling. scale bars: 1 mm. Lower rows: representative microscopic images of coelomic cells used to diagnose each stage. Whereas immature animals display essentially only eleocytes (left), premature animals (middle) show eleocytes as well as germ cells (oocytes and spermatogonia, respectively). In mature animals (right), oocytes have reached their full size of ~180 µm in females, and male spermatogonia have differentiated into spermatozoa. Scale bars: 50 µm. (c) Scheme illustrating the two distinct circalunar phases used for sampling; NM = New Moon; FRFM = Free Running Full Moon (*Zantke et al., 2014*; *Zantke et al., 2013*). (d) Schematic flow chart of the newly developed protocol, allowing to profile the proteome and corresponding transcriptome of each sample.

DOI: https://doi.org/10.7554/eLife.41556.003

The following source data and figure supplements are available for figure 1:

**Source data 1.** Sample IDs.

DOI: https://doi.org/10.7554/eLife.41556.006

**Source data 2.** Differential expression comparisons.

DOI: https://doi.org/10.7554/eLife.41556.007

**Figure supplement 1.** Schematic of representation of the strategy for the generation of the maturation, sex and lunar transcript/proteome regulated lists.

DOI: https://doi.org/10.7554/eLife.41556.004

**Figure supplement 2.** Ambient light conditions and overview over the multiplexing strategies employed for transcriptomic and proteomic analyses.

DOI: https://doi.org/10.7554/eLife.41556.005

Studies from established molecular model systems have shown that sexual behaviour involves neurons expressing sexually-dimorphic markers, including *fruitless* and *doublesex* in *Drosophila*, and *mab* genes in *Caenorhabditis* (e.g. *Kopp, 2012*). In *Platynereis*, differential expression of neuropeptides in whole mature males and females suggests that sexual dimorphism at the level of hormonal regulation coincides with sexual maturation (*Conzelmann et al., 2013*). Given the roles of such neurohormones in other animals, it is likely that they act both locally within the brain, as well as in the periphery of the animals to orchestrate sexual dimorphism of the gonads and behaviour as the animals mature. However, no systematic information is available on sexual dimorphisms in the brain of *Platynereis*.

Importantly, the bristleworm brain also plays a crucial role in circadian and circalunar timing (*Hauenschild, 1956a*; *Hauenschild, 1956b*; *Hauenschild, 1959*; *Hauenschild, 1960*; *Hauenschild, 1966*; *Hofmann, 1976*; *Hofmann and Schiedges, 1984*; *Schenk et al., 2016*; *Zantke et al., 2013*). Its importance in reproductive timing is highlighted by the fact that tails of decapitated animals undergo sexual maturation within two weeks, but are no longer synchronised to the lunar phase (*Hauenschild, 1966*; *Hauenschild, 1974*; *Schenk et al., 2016*). The brain also possesses non-visual photoreceptors that likely receive light inputs for circalunar (and circadian) clock entrainment (*Arendt et al., 2004*; *Tessmar-Raible et al., 2007*; *Zantke et al., 2013*).

Taken together, these findings argue that the *Platynereis* brain integrates information on maturation, sexual differentiation, and circalunar phase in order to bring about lunar-synchronised reproduction. We therefore sought to identify the molecular signatures of these three major biological processes using an integrated molecular profiling approach, and to assess whether a circalunar component could be disentangled from the changes brought about by maturation and sexual differentiation. We carried out this analysis on two levels: First, we systematically assessed transcriptome signatures, building on our previous observation that the transcript levels of a subset of core circadian clock genes are influenced by the phase of the circalunar oscillator in *Platynereis*, a finding that has also been observed in corals and the reef fish *Siganus guttatus* (*Raible et al., 2017*; *Altincicek and Vilcinskas, 2007* references cited therein). Second, we extended our analyses to the proteome level, reasoning that a combined quantitative transcriptomics and proteomics approach would allow us to obtain a better understanding of the major molecular changes that occur on different functional levels, and highlight molecules and pathways that are most prominent in the different processes.

This systematic approach revealed that maturation induces the largest amount of changes on both transcriptome and proteome levels, whereas the circalunar clock has a much stronger impact on the proteome than the transcriptome. Our analyses identified molecular functions, pathways and candidate genes that are diagnostic for either sex and/or maturation stage, and show that representative genes associated with these processes localise in distinct brain regions. We further identify

new factors that are subject to circalunar clock-dependent changes on mRNA and/or protein level, and map their locations in the adult brain. Specifically, we demonstrate that Ependymin-related proteins (ERPs) are strongly affected on both transcript and protein levels by maturation and circalunar clock phases. We further show that the neuropeptide Whitnin/Proctolin, which recently has been correlated with the spawning event in the semilunar spawner abalone *Haliotis asinina* (*York et al., 2012*), is also regulated by the circalunar clock in *Platynereis*.

Taken together, our work provides a comprehensive co-evaluation of transcriptomic and proteomic regulation of stage, sex, and circalunar phase-matched tissue samples, and provides the first systematic insight into the distinct genes and molecular processes influenced by maturation, sexual differentiation and the circalunar clock in adult *Platynereis dumerilii*.

## Results

In order to gain first systematic insight into the molecular impact of the circalunar clock in the brain, we conducted a comprehensive analysis of changes in the *Platynereis* head at two distinct phases of the circalunar clock: new moon (NM) and free-running full moon (FRFM). The environmental conditions are the same at both circalunar phases, FRFM describes the chronobiological conditions in which the animals would normally be presented with a nocturnal light stimulus ('full moon'- FM), which is however not given (compare *Figure 1a* vs. c). Thus, their endogenous circalunar clock is under free-running conditions (i.e. FRFM, see also *Zantke et al., 2013*). In order to disentangle these respective circalunar molecular changes from those primarily related to maturation or sexual differentiation, we performed this analysis separately for male and female worms, across three stages of maturation (*Figure 1b*, *Figure 1—figure supplement 1*). We started our analysis with gene expression profiling by RNA sequencing (RNA-Seq), to assess effects on the transcriptome. As outlined further below, we developed an extraction method that allowed us to subsequently complement this transcriptomic analysis with a comprehensive proteomic analysis on material from the identical samples (*Figure 1d*, *Figure 1—figure supplement 2*).

### A maturation-stage and sex-locked lunar head transcriptome

As the current *Platynereis dumerilii* genome assembly is not yet optimal for unique mapping of short reads for quantitative analyses, we generated a mapping resource based on transcript data. Whereas previous *Platynereis* transcriptomes were generated from whole animals during various adult life stages (*Conzelmann et al., 2013*), or early developmental stages (*Chou et al., 2016*), we reasoned that this might result in under-representation of head-specific transcripts. Moreover, when using whole animal samples, the increase in germ cells during maturation (see e.g. *Schenk and Hoeger, 2010*; *Schenk et al., 2016* and references cited therein) could obscure low abundance expression, which might be of regulatory importance.

Therefore, we generated a head reference transcriptome assembled from RNA-Seq data combined from heads sampled under various circalunar conditions and maturation stages (see Materials and methods). The FRFM condition refers to animals kept in a standard daily 16:8 hr light dark (LD) light cycle whose circalunar clock was previously entrained by nocturnal light for at least two months, but are not exposed to nocturnal light at the expected full moon time when sampling was performed. Such 'free-running' conditions can thus reveal effects solely caused by the endogenous circalunar clock (*Zantke et al., 2013* and *Figure 1c*).

Our primary assembly (Pdu_HeadRef_TS_v1) was systematically curated to (i) remove redundancies, (ii) merge scaffold information provided by published *Platynereis* sequences, (iii) remove sequence contaminations (e.g. prokaryotic), and (iv) flag additional sequences with high similarity to other species (see Materials and methods section), resulting in our final reference sets (Pdu_HeadRef_TS_v4 and v5, see *Figure 2*, *Figure 2—source data 1–3* and Materials and methods section).

To identify gene sets that were regulated by either one of the three sampled parameters (circalunar clock phase, maturation, and sex), and assess to which extent these gene sets were distinct, we performed differential expression analyses using DESeq2 (v1.10.0, *Love et al., 2014*) and EdgeR (v3.12.0, *Robinson et al., 2010*) (*Figure 3a, b*, *Figure 3—figure supplements 1–9*, *Figure 1—figure supplement 1*, *Figure 1—source datas 1* and *2*, *Figure 3—source datas 1* and *2*). Transcripts passing a Benjamin-Hochberg (BH) false discovery rate (FDR) of 10% were considered as significant (FDR, *Benjamini and Hochberg, 1995*). To compare the results of both methods, rank sum files

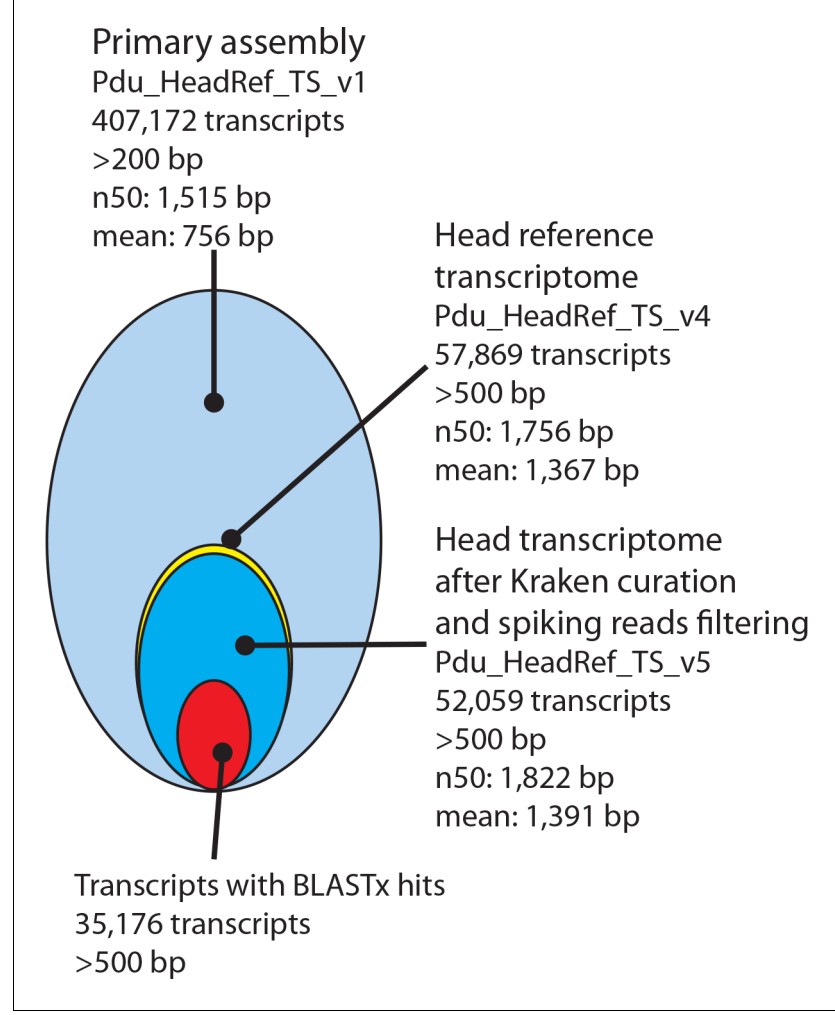

**Figure 2.** Processing and basic features of the *Platynereis* head transcriptome. The primary assembly consisting of 407,172 transcripts with >200 bp was successively processed by: i) collapsing multi-copy transcripts to the longest transcripts of each cluster, ii) removing transcripts shorter than 500 bp, ii) removing transcripts arising from common model organisms or potential food sources, iv) substituting less complete transcripts with previously published sequences from NCBI and the transcriptome published by *Conzelmann et al. (2013)*, and v) adding missing sequences from NCBI. In total 35,176 Pdu_HeadRef_TS_v5 sequences have BLASTx hits against the NCBI nr- database.

DOI: https://doi.org/10.7554/eLife.41556.008

The following source data is available for figure 2:

**Source data 1.** Excluded transcripts.
DOI: https://doi.org/10.7554/eLife.41556.009

**Source data 2.** Platynereis head reference transcriptome v1 base annotation.
DOI: https://doi.org/10.7554/eLife.41556.010

**Source data 3.** Platynereis head reference transcriptome v4 interpro annotation.
DOI: https://doi.org/10.7554/eLife.41556.011

**Source data 4.** Transcriptome ENA accession numbers.
DOI: https://doi.org/10.7554/eLife.41556.012

were generated to combine the results of both DESeq2 and EdgeR differential expression testing, to yield the final lists of differentially expressed transcripts (DETs) for each condition (*Figure 3— source datas 3*, *4*, *5*, *9*, *10* and *11*). Of the three major comparison groups, that is maturation, sex biased and circalunar, maturation exhibits the largest number of DETs (*Altincicek and Vilcinskas, 2007*); 16.52% of all transcripts, *Figure 3a*). Sexual differentiation was characterised by 640 DETs

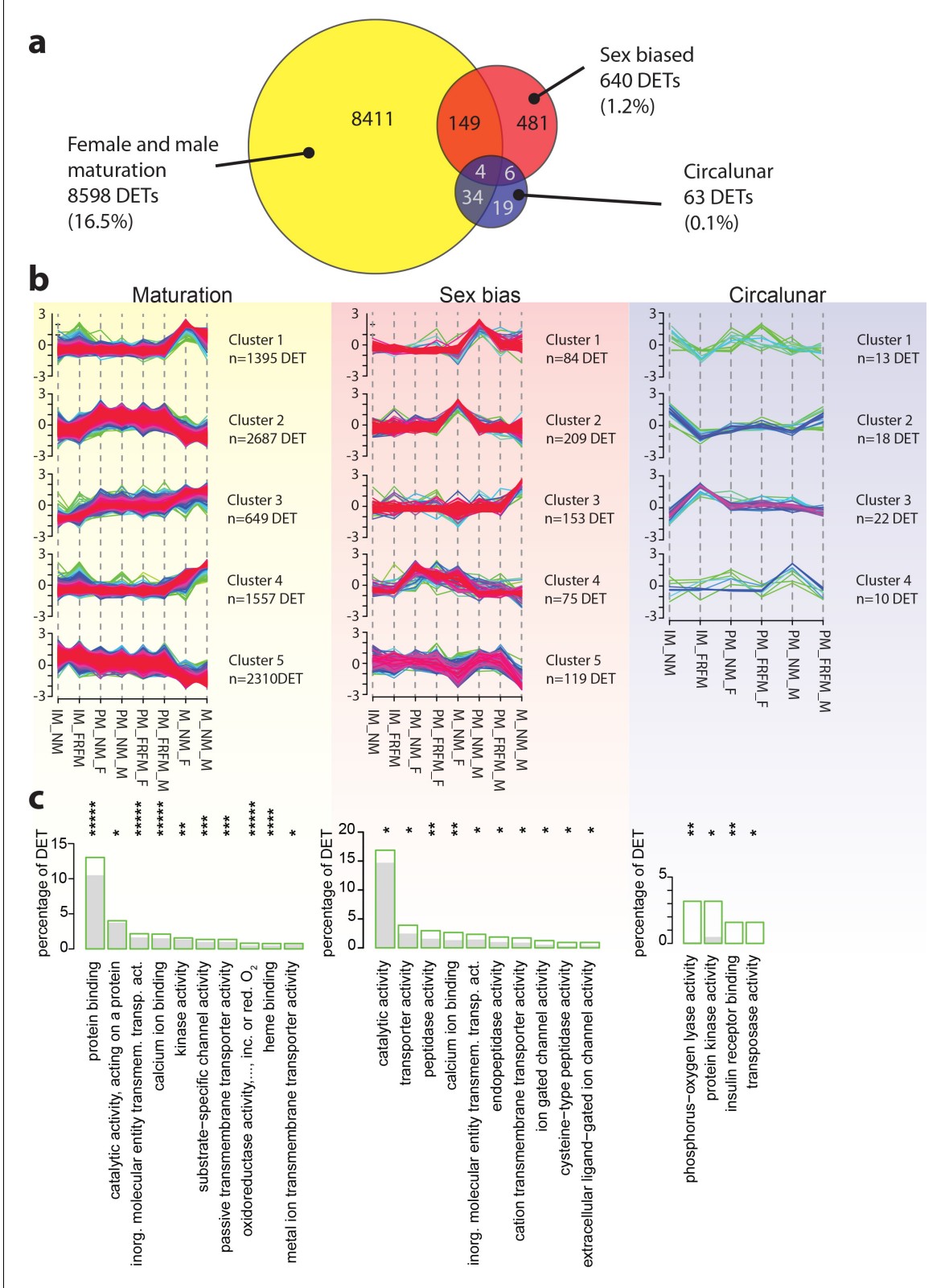

**Figure 3.** Maturation, sex and circalunar phase have distinct transcriptomic signatures in *Platynereis* heads. (a) Venn diagram showing the overall numbers of genes identified as significantly regulated within and across the three different biological processes. Percent in brackets indicate the respective fraction compared to total number of analyzed IDs. (b) Soft clustering of differentially expressed genes using the Mfuzz algorithm. Maturation (yellow) and sexual differences (light red) yielded five clusters; circalunar phase-regulated (light blue) expression was best represented by

*Figure 3 continued on next page*

*Figure 3 continued*

four clusters. Compare to *Figure 5b* for similarity to regulated protein clusters. (c) Results from the GO-term enrichment analysis using the GOStats package on the differentially expressed genes in each comparison; the ten most abundant terms in the Molecular Function category are displayed (for more GO-results see *Figure 3—figure supplement 1–9*). Green boxes: actual percentage of differentially expressed transcripts for each term; grey bars: expected number of transcripts per category. Maturation: protein binding: GO:0005515, catalytic activity, acting on a protein: GO:0140096, inorganic molecular entity transmembrane transporter activity: GO:0015318, calcium ion binding: GO:0005509, kinase activity: GO:0016301, substrate-specific channel activity: GO:0022838, passive transmembrane transporter activity: GO:0022803, oxidoreductase activity, acting on paired donors, with incorporation or reduction of molecular oxygen: GO:0016705, heme binding: GO:0020037, metal ion transmembrane transporter activity: GO:0046873. Sex bias: catalytic activity: GO:0003824, transporter activity: GO:0005215, peptidase activity: GO:0008233, calcium ion binding: GO:0005509, inorganic molecular entity transmembrane transporter activity: GO:0015318, endopeptidase activity: GO:0004175, cation transmembrane transporter activity: GO:0008324, ion gated channel activity: GO:0022839, cysteine-type peptidase activity: GO:0008234. Circalunar phase: phosphorus-oxygen lyase activity: GO:0016849, protein kinase activity: GO:0004672, insulin receptor binding: GO:0005158, transposase activity: GO:0004803. Statistical significance was tested with a hypergeometric G-test implemented in the GOStats package. *p < 0.05, **p < 0.01, ***p < 0.001, ****p < 0.0001, *****p < 0.00001.

DOI: https://doi.org/10.7554/eLife.41556.013

The following source data and figure supplements are available for figure 3:

**Source data 1.** *Platynereis* head reference transcriptome v5 read counts.
DOI: https://doi.org/10.7554/eLife.41556.024

**Source data 2.** Transcriptome: DESeq normalised read counts.
DOI: https://doi.org/10.7554/eLife.41556.025

**Source data 3.** *Platynereis* head reference transcriptome v5 gene universe.
DOI: https://doi.org/10.7554/eLife.41556.026

**Source data 4.** Transcriptome: DESeq output tables.
DOI: https://doi.org/10.7554/eLife.41556.027

**Source data 5.** Transcriptome: EdgeR output tables.
DOI: https://doi.org/10.7554/eLife.41556.028

**Source data 6.** Transcriptome: DESeq2-EdgeR rank sum files.
DOI: https://doi.org/10.7554/eLife.41556.029

**Source data 7.** Transcriptome: Mfuzz cluster gene IDs with cluster membership values.
DOI: https://doi.org/10.7554/eLife.41556.030

**Source data 8.** Transcriptome: all over- and under-represented GO-Terms from all comparisions.
DOI: https://doi.org/10.7554/eLife.41556.031

**Source data 9.** Transcriptome: over- and under-represented GO-Terms maturation comparison.
DOI: https://doi.org/10.7554/eLife.41556.032

**Source data 10.** Transcriptome: over- and under-represented GO-Terms sex bias comparison.
DOI: https://doi.org/10.7554/eLife.41556.033

**Source data 11.** Transcriptome: over- and under-represented GO-Terms circalunar comparison.
DOI: https://doi.org/10.7554/eLife.41556.034

**Source data 12.** Transcriptome: over-represented GO-Term unique for any of the three comparisons.
DOI: https://doi.org/10.7554/eLife.41556.035

**Source data 13.** All transcripts significantly regulated during maturation (maturation DETs).
DOI: https://doi.org/10.7554/eLife.41556.036

**Source data 14.** All transcripts significantly regulated during sexual differentiation (sex DETs).
DOI: https://doi.org/10.7554/eLife.41556.037

**Source data 15.** All transcripts significantly regulated between circalunar phases (lunar DETs).
DOI: https://doi.org/10.7554/eLife.41556.038

**Figure supplement 1.** Analyses of over-represented GO-terms support the existence of distinct molecular signatures for maturation, sex and circalunar clock phase.
DOI: https://doi.org/10.7554/eLife.41556.014

**Figure supplement 2.** Analyses of under-represented GO-terms support the existence of distinct molecular signatures for maturation, sex and circalunar clock phase.
DOI: https://doi.org/10.7554/eLife.41556.015

**Figure supplement 3.** Heat map displaying under-represented GO-terms in all categories.
DOI: https://doi.org/10.7554/eLife.41556.016

**Figure supplement 4.** Heat map displaying under-represented GO-terms in the Biological Process category.
DOI: https://doi.org/10.7554/eLife.41556.017

**Figure supplement 5.** Heat map displaying under—represented GO-terms in the Cellular Compartment category.
DOI: https://doi.org/10.7554/eLife.41556.018

*Figure 3 continued*

**Figure supplement 6.** Heat map displaying under-represented GO-terms in the Molecular Function category.
DOI: https://doi.org/10.7554/eLife.41556.019
**Figure supplement 7.** Heat map displaying over-represented GO-terms in all categories.
DOI: https://doi.org/10.7554/eLife.41556.020
**Figure supplement 8.** Heat map displaying over-represented GO-terms in the Biologcal Process categogy.
DOI: https://doi.org/10.7554/eLife.41556.021
**Figure supplement 9.** Heat map displaying over-represented GO-terms in the Cellular Compartment category.
DOI: https://doi.org/10.7554/eLife.41556.022
**Figure supplement 10.** Heat map displaying over-represented GO-terms in the Molecular Function category.
DOI: https://doi.org/10.7554/eLife.41556.023

(1.23%), whereas 63 DETs (0.12%) appear to distinguish the two sampled phases of the circalunar clock. Of the 640 sex-biased DETs, 149 (23.28%) were also regulated by maturation and six (0.94%) by the circalunar phase (*Figure 3—source data 3–5*); of the 63 circalunar DETs 34 (53.97%) were also regulated by maturation. Four transcripts were regulated in all three parameters (*Figure 3a*).

To further validate our quantitative sequencing results, we selected transcripts from the top 50 ranked DETs for additional qRT-PCR analyses, preferring those that also had evidence on the protein level (see below). These experiments showed that the differences observed in our sequencing approach generally validated well when assessed by qRT-PCR on independent samples (*Figure 4a–d*, *Figure 4—figure supplement 1*). Within these experiments, the tested circalunar candidates appeared more variable (*Figure 4—figure supplement 1*). We noted that the majority of circalunar candidate DETs exhibited generally lower overall expression levels (*Figure 4—figure supplement 1*) and showed additional susceptibility to strain specific polymorphisms or phase differences in their circalunar rhythm profiles, similar to previous findings (*Zantke et al., 2014*).

## The *Platynereis* head transcriptome reveals clear distinctions between maturation, sexual differentiation and circalunar rhythms

After confirming the overall fidelity of the sequencing experiment, we next investigated whether the specific differentially expressed gene sets fell into different regulatory subsets. For this, we analysed the expression profiles by soft clustering analysis using the Mfuzz R-package (*Futschik and Carlisle, 2005*; *Kumar and Futschik, 2007*). Mfuzz clustering on the mean expression values (details see Materials and methods) resulted in five clusters each for maturation- and sex-related DETs, and four for the circalunar DETs (*Figure 3b*, gene lists *Figure 3—source data 6*). The maturation-related DETs showed two clusters with transcripts peaking in the premature (PM) stage, two with a peak in the mature (M) stage, and one in which the expression values declined from immature (IM) over PM to M. The two most prominent clusters, with 2,687 DETs and 2,310 DETs, respectively, were those in which expression was reduced in mature animals (clusters 3 and 5 in *Figure 3b*), while the single cluster showing an increase in transcript levels from IM over PM to M contained the least number of transcripts (649 DETs, cluster 3).

Clustering of the sex-related DETs set yielded four clusters with different profiles. Two of these displayed peaks of transcript levels either in mature females or mature males (clusters 2 and 3, respectively), whereas the other two clusters harboured transcripts peaking either in PM and M females, or in PM and M males (clusters 4 and 1, respectively, see *Figure 3b*). Cluster one was less clear, exhibiting a trough of expression in mature animals of either sex.

Due to the comparatively low number of DETs in the circalunar comparison, the patterns of the circalunar-related DET clusters were less clear than for the sex and maturation related DETs. However, transcripts belonging to clusters two and three containing 22 and 18 DET of the 63DET, respectively, clearly exhibited differences between IM_FRFM compared to IM_NM, specifically a peak in IM_FRFM (cluster three) and a trough in clusters one and two (*Figure 3b*).

Taken together, these results suggest that the three assayed conditions – maturation, sex, and circalunar clock phase – affected largely different sets of genes and by this delineate specific brain 'molecular signatures'. To further analyse if the individual signatures also differed with respect to functional or molecular categories, Gene Ontology annotations were analysed using the GOStats R-package (*Falcon and Gentleman, 2007*).

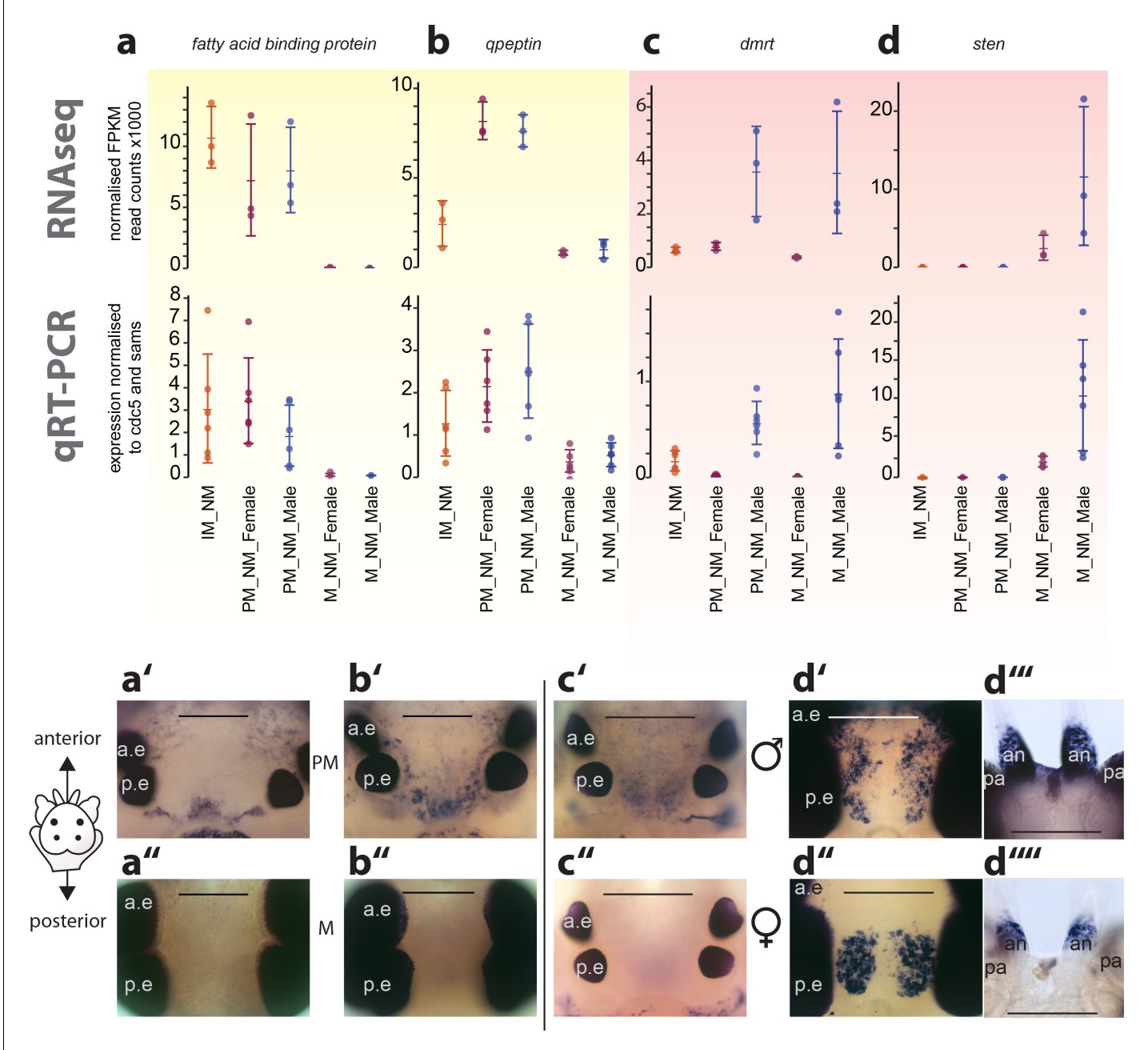

**Figure 4.** Exemplary validation of candidate genes by qRT-PCR and expression domains in the head. (a–a'',b–b'') Maturation markers *fatty acid binding protein* and *qpeptin*. (c–c'',d–d'') Sexual differentiation markers *dmrt* and *sten*. (a–d): qRT-PCR validation of the candidates relative to the arithmetic mean of the reference genes *cdc5* and *sams*. All graphs show the arithmetic mean with standard deviation, and the individual data points. (a'–d'') Whole mount in situ hybridisations of the respective genes, comparing premature animals (PM, (a',b') with mature animals (M, a'',b''), and males (c',d',d''') with females (c'',d'', d''), respectively. All images except d'''and d''': dorsal views, anterior to the top, scale bar: 250 μm. d''' and d'''': ventral views, anterior top, scale bar: 200 μm. Also see *Figure 4—figure supplement 1* for further validations. a.e.: anterior eye; p.e.: posterior eye, an: antenna, pa: palpa.

DOI: https://doi.org/10.7554/eLife.41556.039

The following source data and figure supplement are available for figure 4:

**Source data 1.** Expression values relative to *sams* and *cdc5* as obtained by qRT-PCR.

DOI: https://doi.org/10.7554/eLife.41556.041

**Figure supplement 1.** Exemplary validation of additional transcripts differentially expressed during maturation, between sexes or between circalunar clock phases:

*Figure 4 continued on next page*

*Figure 4 continued*

DOI: https://doi.org/10.7554/eLife.41556.040

These analyses revealed that overall 239 out of a total 4,775 GO-terms were significantly enriched in our comparisons (*Figure 3c*; *Figure 3—figure supplements 1–9*, *Figure 3—source data 7–11*). When analysing the ten most abundant GO-terms, these confirmed the above notion of distinct signature processes for maturation, sexual differences and circalunar phase. Specifically, in the Molecular Function category, 'Protein binding' is the most abundant, significantly-enriched single term for maturation, whereas 'catalytic activity' and 'phosphorous-oxygen lyase activity' are most enriched for the sex-specific and circalunar expression comparisons, respectively (*Figure 3c*). The Biological Processes and Cellular Compartment ontologies also showed pronounced differences in significantly over- and under-represented GO-terms (*Figure 3—figure supplements 1–9*, *Figure 3—source datas 6* and *7*). We found little overlap between all the 239 significantly enriched GO-terms when sorted by our three analysis categories: 131 out of 142 were uniquely present in maturation DETs, 83 of 97 were unique to sex differences and ten of fifteen were specific for the circalunar phase DETs (see *Figure 3—figure supplements 2–9* and *Figure 3—source data 11*). No GO-terms were shared between maturation and circalunar DETs. Taken together, these results suggest that maturation, sexual differentiation and circalunar timing – while being tightly coordinated processes - can be clearly distinguished on a molecular level and are likely driven by different biological mechanisms.

The view that these processes can be well separated in *Platynereis* is further supported by the exemplary whole mount in situ hybridisations (ISH): Maturation candidates tested by in situ hybridisation generally showed broadly-distributed expression over the whole brain. One prominent region was found around the posterior eye pair, where two kidney shaped regions of expression were visible, and another in the medial brain between the eyes (*Figure 4a',b',a'',b''*). Sex biased transcripts exhibited similarly broad expression domains, with noted dimorphism of expression domains, and/or staining intensity reflective of expression level differences between males and females (*Figure 4c', d',c'',d'', d''', d''''*).

## Establishment of a combined transcriptomic and proteomic profiling procedure

While transcriptomic analyses provide broad-scale insights into molecular dynamics that occur on the level of mRNA regulation (transcription itself and transcript stability regulation), most physiological processes are driven by protein function, but as has been pointed out, the knowledge about the proteome changes in lunar spawning cycles is missing (*Zoccola and Tambutté, 2015*) We therefore complemented our transcriptomic study with a systematic proteomic analysis. This objective posed two challenges: (i) whereas general transcript analyses have been previously performed for *Platynereis* (*Chou et al., 2016*; *Conzelmann et al., 2013*), no comparable reference set exists yet at proteome level; (ii) we anticipated that maximal comparability between transcriptomic and proteomic data would require a method that was able to isolate both RNA and protein from the same specimen.

To minimise sampling bias and to facilitate comparisons between the transcriptome and proteome data, we took advantage of the different chemical properties of ribonucleic acid and proteins, and the fact that most of the protein is expected to elute as the flow-through from RNA extractions that use $SiO_2$-based columns under denaturing conditions.

However, in these flow-through fractions, proteins are highly diluted and we found that the high concentrations of guanidin thiocyanate (GuSCN) impair downstream analyses (data not shown). Thus, we first established an appropriate method for concentrating the protein fraction. We freeze-dried and then reconstituted the sample in the original volume, followed by protein precipitation. Due to the presence of thiocyanate in the samples acid based precipitations were not feasible. We tested different alcohol-based precipitation protocols (details see Materials and methods), and found that a 5:1 dilution with acetone yielded the best recovery. The resulting pellets were washed thoroughly in ice-cold EtOH to remove any traces of GuSCN which may damage the filters (FASP protocol, *Wiśniewski et al., 2009*). We found no evidence for non-precipitated proteins in the

acetone precipitation supernatant (for further details see Materials and methods) encouraging us to continue with quantitative proteomic analyses.

To enable quantitative proteomic analysis, open reading frames (ORFs) > 100 amino acids were predicted for all transcripts in the HeadRef_Tsv4 transcriptome for all six possible reading frames, resulting in 92,645 predicted protein sequences (45,065 transcripts; Pdu_HeadRef_prot_v4). In total we identified 2,290 proteins with at least two unique peptides identified per protein from our experimental samples. The number of identified proteins with these stringent criteria matches well with analyses from other invertebrate head/brain proteomes; 2,987 proteins were identified in *Apis mellifera* (*Hernández et al., 2012*) and 3,004 proteins in the larval brain of *Bombyx mori* (*Li et al., 2016*), despite the fact that both of these studies employed an unlabelled shotgun proteomics approach and required only one unique peptide match per protein.

To further compare to existing datasets, we correlated the normalised log10-transformed transcriptome and proteome expression values to each other. This showed a significant, but relatively low correlation (cor.test()) Pearson-correlation coefficient of 0.482 (*Figure 5—figure supplement 1*). This relatively low correlation between transcript and protein levels is highly consistent with previous transcriptome-proteome comparisons studies. Correlations with very similar R-values were found for the developmental transcriptome/proteome in *Drosophila melanogaster* (*Casas-Vila et al., 2017*) and sexual difference transcriptome/proteome comparisons in *C.elegans* (*Tops et al., 2010*). A comparison of protein and mRNA expression levels for multiple stages of *C.elegans* development resulted in even much lower R values (*Grün et al., 2014*). These (and further) studies imply that only limited conclusions about the level of protein can be drawn based on its transcript level in an organism, further emphasising the importance of proteomic comparisons. The consistency with other studies provides further evidence that our methodological approach did not negatively impact on the proteome data quality.

As final quality estimation, we took advantage of another *Platynereis* head proteome that was independently generated using a conventional sample processing method, a label-free approach (see Materials and methods) and did not include large-scale comparisons between different maturation- or sex-specific stages. In total, 3,847 proteins were detectable in this conventionally-processed (see *Supplementary file 1*), label-free proteome set, out of which 2,105 proteins overlapped with the transcriptome/proteome co-processed proteome. To test for possible large-scale systematic compositional biases caused by the different sample processing, we performed a GO-term analysis with both sets comparing to all possible annotated GO-terms and relative frequencies present in the transcriptome dataset (taking all three categories - molecular process, cellular compartment and function - into account). In the 2,290 proteins detected in the transcriptome/proteome co-processed set, 372 significantly over- and 244 under-represented GO-terms were present; whereas in the conventionally-processed set (3,847 proteins) 444 G0-terms were over- and 248 under-represented (see *Figure 5—source datas 3* and *4*). Overall, the types of over- and under-represented GO-terms were similar and, in the majority, overlapped (*Figure 5—source data 3–5*), suggesting that the transcriptome/proteome co-processing of our samples did not induce any significant bias in terms of functional complexity.

## A maturation-stage and sex locked circalunar head proteome

We next analysed the proteome with respect to the different sampling conditions. To enable robust comparison between all conditions, we defined a subset of 'quantifiable protein' comprising only those detected uniquely in at least 2 BR and in 3 out of the 5 TR per BR for all samples, yielding a total of 1,064 proteins; a number comparable to studies of the mosquito *Aedes aegypti* head proteome, applying similar stringency filters (1,139 proteins, *Nunes et al., 2016*). We tested for significantly different protein abundances using ROTS and LIMMA (*Elo et al., 2008*; *Elo et al., 2009*; *Ritchie et al., 2015*) statistics and combined the results from both tests using the same rank sum approach used previously for the transcriptome DET analyses (*Figure 5—source datas 6–8* and *19–21*). As for the transcriptome data, the largest proportion of proteins, that is 693 proteins were differentially expressed over the process of maturation, corresponding to 65.13% of quantifiable proteins (*Figure 5a*). Contrary to the transcriptome profiling results, however, sexual differentiation accounted for only 17 differentially expressed proteins (DEP, 1.60%). Conversely, circalunar phase affected 261 DEP (24.53%), significantly exceeding the amount observed in the transcriptomic

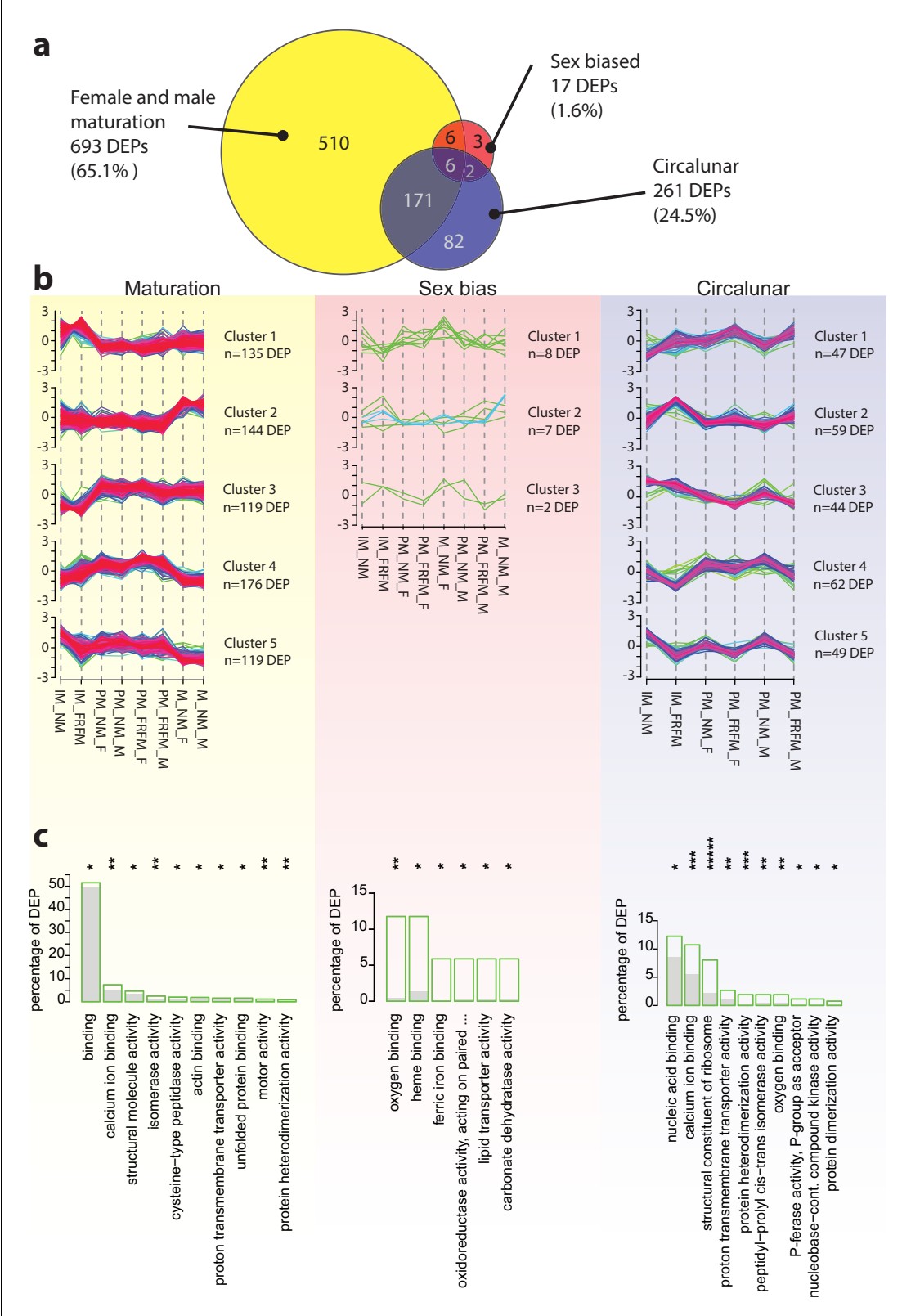

**Figure 5.** The *Platynereis* head proteome is differentially affected by maturation, sex and circalunar phase. (a) Venn diagram showing the proteins significantly regulated within and across the three different biological processes in the three major comparisons made (brackets indicate respective fraction of total comparable proteins) (b) Soft clustering of differentially expressed proteins using the Mfuzz algorithm. The expression data for maturation and circalunar phase are best represented by five and for sexually dimorphic protein expression by three clusters. Compare to *Figure 3b* for
*Figure 5 continued on next page*

*Figure 5 continued*

similar dynamics in the regulated transcript clusters. (c) GO-term enrichment analysis using the GOStats package on the differentially expressed proteins in each comparison showing the ten most abundant terms in the Molecular Function category (further GO-term analyses: see *Figure 5—figure supplement 2–6*). Green boxes show the percentage of differentially expressed proteins for each term; the grey bars depict the expected amount of proteins per category. Maturation: binding: GO:0005488, calcium ion binding: GO:0005509, structural molecule activity: GO:0005198, isomerase activity: GO:0016853, cysteine-type peptidase activity: GO:0008234, actin binding: GO:0003779, proton transmembrane transporter activity: GO:0015078, unfolded protein binding: GO:0051082, motor activity, GO:0003774, protein heterodimerization activity, GO:0046982. Sex bias: oxygen binding: GO:0019825, heme binding: GO:0020037, ferric iron binding: GO:0008199, oxidoreductase activity, acting on paired donors, with incorporation or reduction of molecular oxygen, reduced pteridine as one donor, and incorporation of one atom of oxygen (oxidoreductase activity, acting on paired...): GO:0016714, lipid transporter activity: GO:0005319, carbonate dehydratase activity: GO:0004089. Circalunar phase: nucleic acid binding: GO:0003676, calcium ion binding: GO:0005509, structural constituent of ribosome: GO:0003735, proton transmembrane transporter activity: GO:0015078, protein heterodimerization activity: GO:0046982, peptidyl-prolyl cis-trans isomerase activity: GO:0003755, oxygen binding: GO:0019825, phosphotransferase activity, phosphate group as acceptor: GO:0016776, nucleobase-containing compound kinase activity: GO:0019205, protein dimerization activity: GO:0046983. Statistical significance was tested with a hypergeometric test implemented in the GOStats package. *$p < 0.05$, **$p < 0.01$, ***$p < 0.001$, *****$p < 0.00001$.

DOI: https://doi.org/10.7554/eLife.41556.042

The following source data and figure supplements are available for figure 5:

**Source data 1.** Labelled proteomics: detected proteins and intensities, filtered and unfiltered.
DOI: https://doi.org/10.7554/eLife.41556.050

**Source data 2.** *Platynereis* head reference proteome v4 gene universe.
DOI: https://doi.org/10.7554/eLife.41556.051

**Source data 3.** Proteome comparison: all over- and under-represented GO-Terms labelled proteomics.
DOI: https://doi.org/10.7554/eLife.41556.052

**Source data 4.** Proteome comparison: all over- and under-represented GO-Terms unlabelled proteomics.
DOI: https://doi.org/10.7554/eLife.41556.053

**Source data 5.** Proteome comparision: over- and under-represented GO-Terms shared between labelled and unlabelled proteomics.
DOI: https://doi.org/10.7554/eLife.41556.054

**Source data 6.** Proteome: ROTS results.
DOI: https://doi.org/10.7554/eLife.41556.055

**Source data 7.** Proteome: LIMMA results.
DOI: https://doi.org/10.7554/eLife.41556.056

**Source data 8.** Proteome: ROTS-LIMMA rank sum files.
DOI: https://doi.org/10.7554/eLife.41556.057

**Source data 9.** Proteome: Mfuzz cluster protein IDs with cluster membership values.
DOI: https://doi.org/10.7554/eLife.41556.058

**Source data 10.** Proteome: all over- and under-represented GO-Terms from all comparisions.
DOI: https://doi.org/10.7554/eLife.41556.059

**Source data 11.** Proteome: over- and under-represented GO-Terms maturation comparison.
DOI: https://doi.org/10.7554/eLife.41556.060

**Source data 12.** Proteome: over- and under-represented GO-Terms sex bias comparison.
DOI: https://doi.org/10.7554/eLife.41556.061

**Source data 13.** Proteome: over- and under-represented GO-Terms circalunar comparison.
DOI: https://doi.org/10.7554/eLife.41556.062

**Source data 14.** Lunar proteome repetion: detected proteins and intensities, filtered and unfiltered.
DOI: https://doi.org/10.7554/eLife.41556.063

**Source data 15.** Lunar proteome repetion: ROTS-LIMMA significant proteins.
DOI: https://doi.org/10.7554/eLife.41556.064

**Source data 16.** Lunar proteome set1-set2 ovelapping IDs ROTS-LIMMA significant proteins in set1.
DOI: https://doi.org/10.7554/eLife.41556.065

**Source data 17.** Lunar proteome set1-set2 ovelapping IDs ROTS-LIMMA significant proteins in set2.
DOI: https://doi.org/10.7554/eLife.41556.066

**Source data 18.** Lunar proteome: all ROTS-LIMMA significant proteins with BLASTx annotation.
DOI: https://doi.org/10.7554/eLife.41556.067

**Source data 19.** All proteins significantly regulated during maturation (maturation DEPs).
DOI: https://doi.org/10.7554/eLife.41556.068

**Source data 20.** All proteins significantly regulated during sexual differentiation (sex DEPs).
DOI: https://doi.org/10.7554/eLife.41556.069

*Figure 5 continued on next page*

*Figure 5 continued*

**Source data 21.** All proteins significantly regulated between circalunar phases (lunar DEPs).
DOI: https://doi.org/10.7554/eLife.41556.070
**Figure supplement 1.** Overall correlation of protein and transcript abundances detected in the same biological samples.
DOI: https://doi.org/10.7554/eLife.41556.043
**Figure supplement 2.** Analyses of over-represented GO-terms support the existence of distinct molecular signatures for maturation, sex and circalunar clock phase
DOI: https://doi.org/10.7554/eLife.41556.044
**Figure supplement 3.** Heat map displaying over-represented GO-terms in all categories.
DOI: https://doi.org/10.7554/eLife.41556.045
**Figure supplement 4.** Heat map displaying over-represented GO-terms in the Biological Process Category.
DOI: https://doi.org/10.7554/eLife.41556.046
**Figure supplement 5.** Heat map displaying over-represented GO-terms in the Cellular Compartment Category.
DOI: https://doi.org/10.7554/eLife.41556.047
**Figure supplement 6.** Heat map displaying over-represented GO-terms in the Cellular Compartment Category.
DOI: https://doi.org/10.7554/eLife.41556.048
**Figure supplement 7.** The normalisation procedure applied to the raw protein abundance values leads to evenly spread and centred abundance profiles.
DOI: https://doi.org/10.7554/eLife.41556.049

analysis. Of the 17 sex biased proteins, 12 were regulated during maturation, as were 177 of the 261 (67.82%) circalunar proteins. Six proteins were affected by all three processes (*Figure 5a*).

We are aware that transcriptomic versus proteomic comparisons are of course different in that RNA sequencing will allow to quantify most transcript types present in a given sample, whereas quantitative proteomics can at present only measure the most abundant proteins present in this sample. This likely explains the difference in the number of sex-specific DETs vs. DEPs, but not the difference observed in the numbers of circalunar DETs vs. DEPs.

To further assimilate the proteomic data with the transcriptomic profiles, we performed the same type of soft cluster analysis of the significant DEPs using the Mfuzz algorithm (*Futschik and Carlisle, 2005*; *Kumar and Futschik, 2007*) to assess if co-regulated clusters existed for the respective processes. Maturation as well as circalunar DEPs each clustered robustly into five protein expression clusters, whereas the sex-specific DEPs showed only weak associations to three clusters (*Figure 5b*, gene lists *Figure 5—source data 9*). A comparison between protein and RNA expression clusters for the maturation process revealed similar dynamic groups (*Figure 3b* vs. 5b), however, these were not necessarily populated by corresponding transcripts and proteins, as only 40.12% (278 out of 693) of DETs showed a similar profile of regulation on the protein level. This finding suggests that transcript to protein changes are 'offset' in time for the majority of the transcripts. Comparisons between DET and DEP clusters were less robust for the sex-specific and circalunar categories, given that only few DEPs or DETs were identified, respectively.

GO enrichment analysis of the DEPs showed that out of a total of 2,448 GO terms present in all 2,290 detectable proteins, 121 GO terms were enriched. Of these, 75 (62.5%) GO terms were found to be enriched in maturation DEPs, 17 in sex biased DEPs, and 46 in circalunar DEPs. Like for the transcriptome, there was little overlap between the GO term enrichments across the three different analysed biological processes (*Figure 5c*, *Figure 5—figure supplements 2–6*, *Figure 5—source data 10–13*).

## Additional proteomic analyses confirm the large change in protein abundance between different circalunar phases

In our initial proteomic analysis for differences between NM and FRFM stages we discovered a much higher amount of differentially expressed proteins than we would have expected based on the transcriptome data (0.12% DET vs. 24.53% DEP, *Figures 3a* and *5a*). This difference is specific for the circalunar comparison and not present or opposite for the maturation and sex-biased comparisons, respectively (for further interpretation and discussion- see discussion section below). In order to verify that this result was not caused by sampling, extraction or measurement artefacts we repeated the proteomics analysis using independent samples. We limited this analysis to IM stages, which showed

a large proportion of differences between circalunar phases and are less affected by sexual differentiation and maturation.

This second round (filtered as above) led to a set of 1,671 quantifiable proteins (*Figure 5—source data 14*), out of which 1,017 were also present in the initial (1,064 quantifiable proteins) dataset, thus 95.58% of the proteins initially passing our stringent quality filter criteria again passed these criteria, indicating a high reliability and reproducibility of protein detection for this sample type. Of the 1,671 proteins of the new dataset, 173 (10.35 %, *Figure 5—source data 15*) were significantly differentially expressed. Re-analysing the dataset by using only the 1,017 quantifiable proteins shared between the first and second dataset resulted in 64 DEPs (testing was limited to IM animals, see above) comparable to the 70 DEPs in IM animals in the first analysis (see *Figure 5—source datas 8*, *16* and *17*).

To identify commonly regulated proteins in all tested proteome datasets, irrespective of frequently occurring sequence differences (e.g. due to isoforms or alleles), which could obscure identical ID-calling, we ran BLASTx against the Uniprot DB for all 365 circalunar DEPs identified from both proteomic analyses (i.e. the initial circalunar IM, PM_Female and PM_Male sets and the second IM set) and grouped these according to their BLASTx annotations. We then manually inspected the DEPs for similar identities. Protein groups with similar identities or individual proteins that were either regulated in at least one dataset of the first proteomic analysis and the second proteomic analysis or in all three conditions (IM, PM male, PM female) of the first proteomic analysis (only one – IM - condition was tested in the second proteomic analysis) were marked as candidates.

This resulted in 27 protein groups (i.e. identified proteins with related, but not identical ID) and 29 individual protein IDs, which were repeatedly detected in the different quantitative proteomic comparisons (*Figure 5—source data 18*).

Finally, we tested if the high numbers of circalunar regulated proteins could be explained by the lower number of proteins compared to the high transcriptome number, which reduces the stringency of the Benjamini Hochberg multiple testing algorithm (*Benjamini and Hochberg, 1995*) used to calculate the FDR. We tested how many transcripts are significantly regulated, if we focus the analyses on the top 5,000 or 1,000 expressed transcripts. For the 5,000 most highly expressed transcripts we found 20 circalunar DET (0.40% compared to 0.12% in the complete transcriptome set), a 3.3-fold enrichment; 262 sex different DET (5.24% compared to 1.22% in the complete transcriptome set), a 4.3-fold enrichment and 2,435 maturation regulated DET (48.70% compared to 16.52% in the complete transcriptome set), a 3.3-fold enrichment. For the ,1000 most highly expressed transcripts we found four circalunar DET (0.40% compared to 0.12% in the complete transcriptome set), a 3.3-fold enrichment; 52 sex different DET (5.20% compared to 1.22% in the complete transcriptome set), a 4.3-fold enrichment and 546 maturation regulated DET (54.60% compared to 16.52% in the complete transcriptome set), a 3.3-fold enrichment. This means that statistical differences (due to overall lower numbers tested) can explain the overall increase in the maturation regulated proteins, but cannot explain the about 245-fold increase we observe between circalunar transcriptome and proteome.

## Characterisation of circalunar proteome regulated candidates reveals ependymin-related genes and prepro-whitnin as putative circalunar clock phase signalling molecules

We next started to characterise several circalunar DEP candidates and provide a showcase of five of them. To gain a better understanding as to where these proteins are synthesised and possibly active, we analysed their expression in adult heads. These expression analyses are based on whole mount in situ hybridisations, which we think is justified to do as we were interested in their place of origin.

A larger group that caught our attention consisted of iron binding proteins. We selected two of these for validation: Myohaemerythrin (myoHr), and Somaferritin. MyoHr is a representative of the Haemerythrins, small oxygen-binding non-haem iron proteins present in all phyla except for in Deuterostomes; myoHr in particular appears to be restricted to marine invertebrates (*Coates and Decker, 2017*; *Costa-Paiva et al., 2017*). Ferritins are intracellular iron storage proteins, which additional to their role iron metabolism have been implicated in oxidative stress resistance and immunology (*Altincicek and Vilcinskas, 2007*; *Andrews et al., 1992*; *Pham and Winzerling, 2010*).

Another circalunar DEP that caught our attention was annotated as Whitnin precursor. Whitnin is the mollusc orthologue of the insect neuropeptide Proctolin (*Veenstra, 2010*, *Figure 7—figure*

*supplement 1*, *Figure 7—source datas 1* and *2*). Of note, *prepro-whitnin* was shown to be differentially regulated on transcript level in the nervous system of abalone *Haliotis asinina* over the course of two weeks. Maximal expression occurred just prior to the semi-circalunar spawning events of this organism (*York et al., 2012*). Whitnin is thus a potential candidate for relaying the circalunar signal to other cells. The last two DEPs we focused on both belong to the large family of Ependymin-related proteins. These are highly sequence-related and could either represent different alleles of the same gene or are the result of a recent gene duplication event (*Figure 6a*, *Figure 6—figure supplement 1*, *Figure 6—source datas 1* and *2*).

Ependymin-related proteins (ERPs) are a group of small extracellular glycoproteins originally identified as one of the most abundant proteins in teleost-fish cerebrospinal fluid (*Shashoua, 1991*; *Suárez-Castillo and García-Arrarás, 2007*). Later, ERPs were discovered in other vertebrates and due to their complete absence from all ecdysozoan genomes and transcriptomes analysed so far, were considered to be vertebrate specific (*Suárez-Castillo and García-Arrarás, 2007*; *Andrews et al., 1992*) references cited therein). However, recently ERPs have been identified in an increasing number of non-ecdysozoan invertebrates, including echinoderms, hemichordates, molluscs, annelids and even cnidarians (*Hall et al., 2017*; *Suárez-Castillo and García-Arrarás, 2007*). A re-occurring feature of ERPs are their multiple group-specific gene duplications (*Hall et al., 2017*; *Suárez-Castillo and García-Arrarás, 2007*). This can be on a higher level - such as the teleost-specific or echinoderm-specific multiplications, but also on the level of species (groups), as pointed out for the *Acanthaster planci* species group (*Hall et al., 2017*), or as we also observed in *Platynereis dumerilii*. We identified 15 ependymin-like sequences in the transcriptome, most of which cluster together in one specific subgroup (*Figure 6a*, *Figure 6—figure supplement 1*, *Figure 6—source datas 1* and *2*). Seven *Pdu*-ERPs were detected in the proteome, all of which were significantly differentially abundant between the two circalunar phases (*Figure 5—source data 18*). Of those, two showed correlated changes between transcript and proteome level (*Figure 6b,b'* and *Figure 6—figure supplement 2*) and were subsequently investigated for their expression patterns.

Both *Pdu-erps* exhibited identical patterns, covering sensory appendages of the head; palpae and peristomial cirri, a large expression domain on the ventral side of the head, as well as an anterior and posterior domain in the dorsal brain (*Figure 6c–e*, *Figure 6—figure supplement 2*). The other investigated circalunar candidates exhibited overall different expression domains (*Figures 6c–e* and *7*): *Pre-pro-whitnin* is expressed in four main regions. Besides expression in single cells of the peristomial cirri, it is expressed in two areas located in the medial forebrain (*Figure 7a',a''*), and one small paired cluster slightly anterior to and below the anterior pair of adult eyes (*Figure 7a'''*).

Similarly, *myohr* was present in the medial brain and to a lesser part around the posterior eyes (*Figure 7b'b''*). Different to *whitnin* and *myohr*, *ferritin* showed a broad expression in the head (*Figure 7c',c''*). The fact that the expression domains are already largely divergent among these few investigated circalunar candidates, suggests that the effect of the circalunar clock is not confined to specific regions of the head or brain.

## Discussion

In this study, we pioneered a combined proteomic and transcriptomic profiling approach to identify coordinated-yet specific changes associated with maturation, sex and circalunar phase in the marine annelid *Platynereis dumerilii*. The analyses profited from our newly established and benchmarked protocol for the isolation and subsequent quantitative analysis of RNA and protein from the same tissue sample, which does not compromise the complexity of the detected proteins. We expect this method to be widely applicable and be particularly useful for studies that are restricted by the amount of samples that can be obtained and analysed.

Using this protocol for *Platynereis dumerilii*, we find that ~ 17 percent of all analysed transcripts and ~66% of the analysed proteins undergo regulation during maturation. This indicates that the maturation process is the biggest driving force for changes on both transcriptome and proteome levels as worms approach the single circalunar-synchronised spawning event of their lifecycle. This finding is consistent with the fact that physiology, metabolism, body morphology and behaviour change dramatically during the maturation phase: Animals change from a benthic, solitary, and feeding life style to a sexually mature, anorexic state that prepares them for their 'nuptial dance', during which germ cells are released. While this process is accompanied by major morphological changes

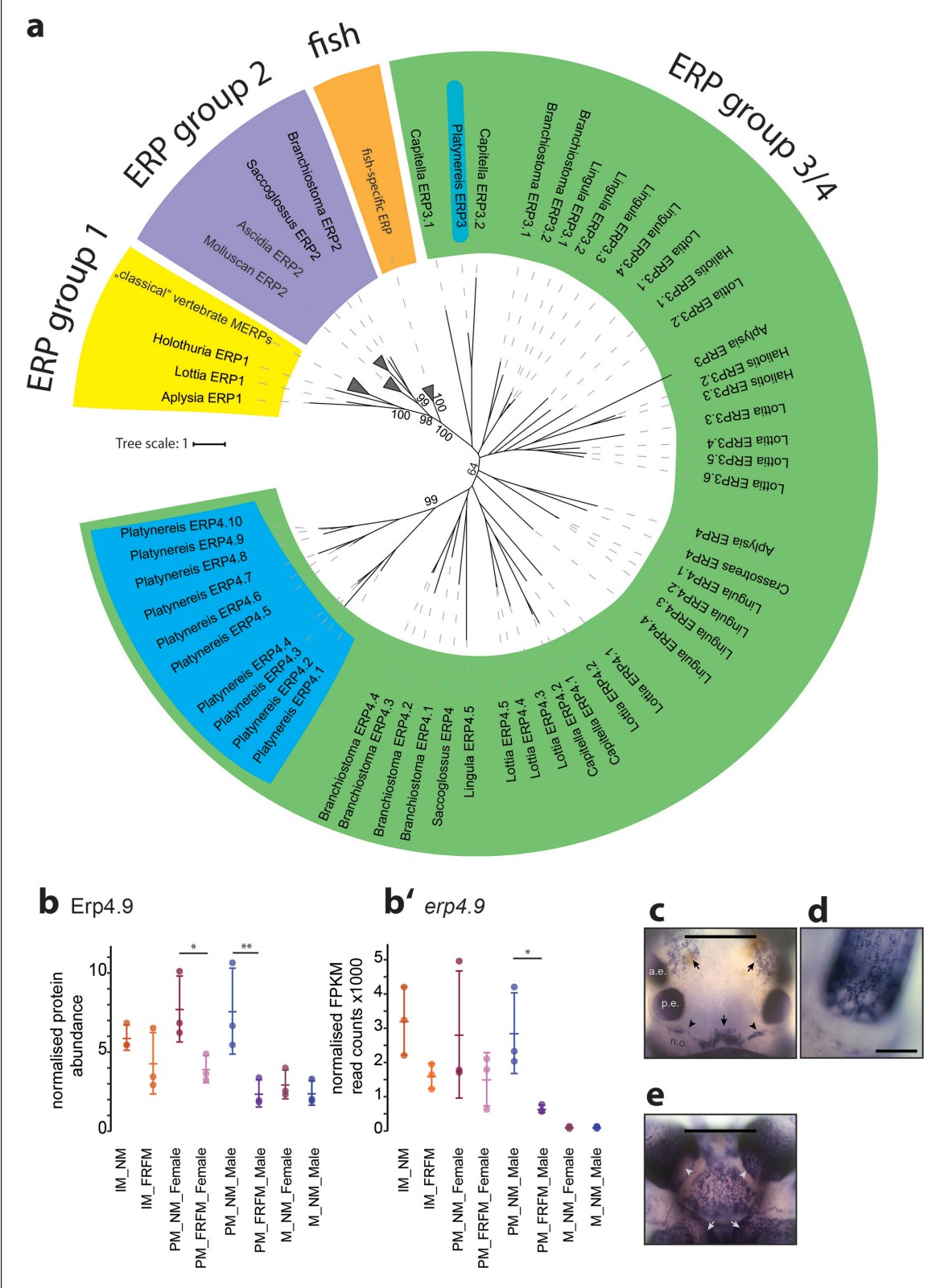

**Figure 6.** Members of a lineage-specific expansion of Ependymin-related proteins are subject to circalunar regulation on RNA and protein level. (a) Unrooted maximum likelihood phyolgenetic tree of Ependymin-related proteins based on the analysis published by *Suárez-Castillo and García-Arrarás, 2007* including eleven of 15 *Platynereis* Ependymin-related proteins (ERP4.1–4.10; ERP3; blue boxes) and various additional invertebrate Ependymin sequences. The phylogeny reveals four distinct groups of Ependymin-related proteins (ERP) that are termed ERP1, ERP2, ERP3/4 and fish-

*Figure 6 continued on next page*

*Figure 6 continued*

specific ERPs; group-specific expansions of ERPs is observed at several places in the phylogeny. All identified *Platynereis* Ependymins (blue) fall into the ERP3/4 clade. A full tree with un-collapsed nodes, the alignment, NCBI accession numbers and the *Platynereis* sequence identifiers are provided in *Figure 6—figure supplement 1* and *Figure 6—source datas 1* and *2*. (b,b') Protein and RNA-Seq expression profiles validate *Platynereis* ERP4.9 as a new target of circalunar phase on RNA and protein level. (b) Normalised protein expression profile; (b') DESeq2 normalised RNA expression profile. The graphs show the arithmetic mean with standard deviation, together with individual data points. (c–e) RNA expression pattern of *erp4.9* in immature adult *Platynereis* heads. a.e.: anterior eye; p.e.: posterior eye; n.o.: nuchal organ. (c) Dorsal view, anterior to the top. Characteristic expression is observed around the dorsal blood vessel (central arrow), anterior of the nuchal organ (arrowheads) and next to the anterior eyes (arrows). Scale bar: 250 μm. (d) Dorsal view of the base of a prostomial cirrus, scale bar: 50 μm. (e) Ventral view, anterior to the top. Expression is observed in a plate-like structure at the bottom of the head, as well as around the mouth opening (arrows) and the palps (arrow heads). Scale bar: 250 μm. For additional circalunar-regulated ERPs see *Figure 6—figure supplement 2* and *Figure 5—source datas 16* and *17*.

DOI: https://doi.org/10.7554/eLife.41556.071

The following source data and figure supplements are available for figure 6:

**Source data 1.** Ependymins alignment.
DOI: https://doi.org/10.7554/eLife.41556.074
**Source data 2.** Ependymins alignment sequence IDs/accession numbers.
DOI: https://doi.org/10.7554/eLife.41556.075
**Figure supplement 1.** Maximum likelihood phylogeny of Ependymin-related proteins using the same alignment and settings as in *Figure 6a*.
DOI: https://doi.org/10.7554/eLife.41556.072
**Figure supplement 2.** ERP4.10 is regulated in both protein and RNA expression by circalunar phase.
DOI: https://doi.org/10.7554/eLife.41556.073

in the trunk (in particular, the generation of germ cells that amass to up to 40% of the body mass, accompanied by a deprivation of up to 60% of longitudinal muscle fibres), the head is known to be a major source of factors that orchestrate this transition (*Hauenschild, 1956a*; *Hauenschild, 1956b*; *Hauenschild, 1966*; *Hofmann, 1976*; *Schenk et al., 2016*). The observed changes in the head transcriptome and proteome strongly support this notion, and the recent identification of a major brain hormone component that regulates maturation onset and progression (*Schenk et al., 2016*) provides an interesting entry point to understand how these significant changes are hormonally regulated and conveyed to the entire body.

Maturation of *Platynereis* is also accompanied by a significant increase of overall eye volume (*Fischer and Brökelmann, 1966*). In line with the described expansion of the outer segments of the eye photoreceptors – that typically harbour the photopigments – we observed a significant increase in transcript abundance of the *Platynereis r-opsin1* gene, a major opsin gene expressed in the adult eye photoreceptors (*Arendt et al., 2002*).

Comparing the observed transcriptome changes to the changes found in other species shows variable similarities, depending on brain regions and species. A similar study of the honeybee brain showed that ~40% of the assessed transcripts exhibited changes associated with maturation, the majority of which (~61%) were associated with caste behaviour (*Whitfield et al., 2003*). Other species show less pronounced changes, such as in the human neocortex, in which about 9% of the transcripts were reported to change during postnatal development (*Kang et al., 2011*); in zebrafish ~20% of the pituitary transcripts were differentially expressed during sexual maturation (*He et al., 2014*), while ~11% of the murine hippocampus transcriptome were differentially expressed during maturation (*Bundy et al., 2017*) and the Mediterranean fruit fly *Ceratitis capitata* exhibited up to 7% of head transcriptome changes during maturation (*Gomulski et al., 2012*).

Overall proteome changes in the brain during development/maturation show a distribution range in vertebrates from 2–8% (*Ori et al., 2015*; *Walther and Mann, 2011*), while 15–17% changed in the larval brain development of *Bombyx mori* and *Apis mellifera*, respectively (*Hernández et al., 2012*; *Li et al., 2016*).

Sexual differences in the brain transcriptome are highly dependent on the species studied. In guppies ~ 13% and in the beetle *Callosobruchus maculatus* ~17% of the brain's transcripts have been described to be sex biased (*Sharma et al., 2014*; *Immonen et al., 2017*); while in birds around ~2% of the brain transcripts (*Naurin et al., 2011*) and human brain ~1.1% of the transcripts show a sex bias (*Kang et al., 2011*).

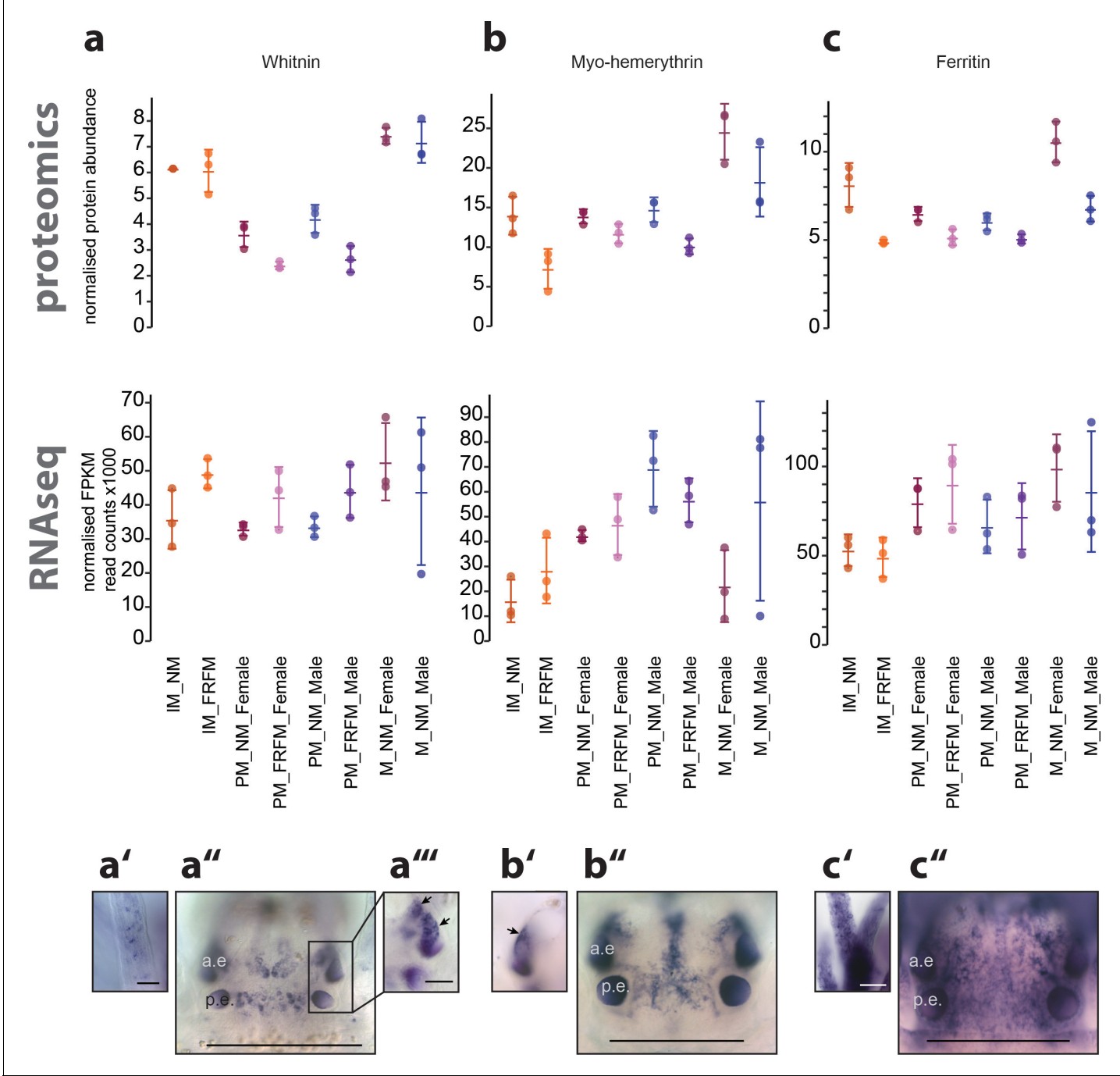

**Figure 7.** Proteome analyses reveal additional molecular markers for circalunar phase with distinct expression profiles and proposed functions. (a–a''') The neuropeptide Whitnin, a'- peristomeal cirrus, a''- head, a'''- indicted head expression in a'' with different focus. (b–b'') The oxygen storage protein Haemerythrin, b''- head, b'- lateral head aspects below the focal plane of b''. (c–c'') The iron storage protein Ferritin. c'- peristomeal cirrus (a–c) compares the normalised protein expression profiles (top, 'proteomics') with the corresponding DESeq2 normalised mRNA expression profile (bottom, 'RNA-Seq'). All graphs show the arithmetic mean with standard deviation, and the individual data points. (a'–c'') Expression domains as characterised by whole-mount in situ hybridisation; (a'',b'',c'') dorsal views of immature *Platynereis* heads stained with riboprobes of the corresponding transcript; All images are oriented with the anterior side to the top; scale bar: 250 µm. a.e.: anterior eye, p.e.: posterior eye.

DOI: https://doi.org/10.7554/eLife.41556.076

The following source data and figure supplement are available for figure 7:

**Source data 1.** Whinin-Proctolin alignment.

DOI: https://doi.org/10.7554/eLife.41556.078

*Figure 7 continued on next page*

*Figure 7 continued*

**Source data 2.** Whitnin-Proctolin alignment sequence IDs/accession numbers.
DOI: https://doi.org/10.7554/eLife.41556.079
**Figure supplement 1.** *Platynereis* Whitnin is part of a Proctolin/Whitnin family of neuropeptides.
DOI: https://doi.org/10.7554/eLife.41556.077

While the observed transcriptome and proteome changes in *Platynereis dumerilii* heads during development, growth and maturation and brain transcriptome changes between different sexes are thus within the range of expectations, we somewhat unexpectedly found that the circalunar phase has little impact on transcript expression changes, but more prominently affects protein level changes. Specifically, we find that only 0.12% of the transcripts we analysed are regulated. This is more than 20-fold less than what has been shown as an impact of the circadian clock on the regulation for brain/head transcriptome in several animals, for example *Aedes aegypti* (6.7%, *Leming et al., 2014*), *Drosophila melanogaster* (2.8%, *Hughes et al., 2012*) and *Mus musculus* (3–4%, *Zhang et al., 2014*). Directly equivalent studies on head/brain proteomes have not been performed, instead specific central nervous system parts of mice and rats have been investigated. For the murine suprachiasmatic nucleus, 13–20% of the identified and quantifiable proteins (871 and 2,112, respectively) exhibited circadian abundance changes (*Chiang et al., 2014*; *Deery et al., 2009*), compared to about 5% of transcriptome changes in the same tissue (*Ueda et al., 2002*). Thus, the proteomic changes we observe between two different circalunar time points (with a similar number of quantifiable proteins) are comparable to the circadian changes observed in the nervous system of other animals.

While it is possible that we underestimate the amount of circalunar transcriptomic changes due to limited sampling resolution (only sampling two circalunar time points and one circadian time point), as well as stringent multiple comparisons statistics (see tests in result section), it is not obvious why such an underestimation effect should be different for the proteome under the same conditions. Alternatively, it is also possible that the level of variability is higher in the transcriptome than in the proteome, which – due to the higher experimental noise - would lead to a lower number of detected transcripts compared to the proteome. However, we do not see evidence for such a scenario in our data.

We therefore suggest as the most parsimonious explanation that we underestimate the occurring circalunar changes in both transcriptome and proteome, but that the observed relative difference is representative. This implies that, differing from the known circadian clock studies, the circalunar oscillator has a significantly stronger impact on the proteome than on the transcriptome. Thus, the mechanisms responsible for circalunar reproductive timing possibly operate and might be coordinated more strongly at post transcriptional, translational/post-translational level(s), than on transcript level(s).

## Materials and methods

**Key resources table**

| Reagent type (species) or resource | Designation | Source or reference | Identifiers | Additional information |
|---|---|---|---|---|
| Strain, strain background (*Platynereis dumerilii*, immature) | IM_NM; IM_FRFM | MFPL Fish and Marine Facility | NA | NA |
| Strain, strain background (*Platynereis dumerilii*, Female) | PM_NM_F; PM_FRFM_F; M_NM_F | MFPL Fish and Marine Facility | NA | NA |
| Strain, strain background (*Platynereis dumerilii*, Male) | PM_NM_M; PM_FRFM_M; M_NM_M | MFPL Fish and Marine Facility | NA | NA |
| Recombinant DNA reagent | pJet1.2 (plasmid) | Thermo Scientific | Thermo Scientific: K1232 | |

*Continued on next page*

*Continued*

| Reagent type (species) or resource | Designation | Source or reference | Identifiers | Additional information |
|---|---|---|---|---|
| Sequence-based reagent | primers see *Supplementary file 2* | this study | | qPCR primers were designed exon-intron spanning using the Roche Universal probe library homepage (see Materials and methods section for details). Primers for cloning were designed based on sequences derived from the transcriptome assembly, the amplicon was aimed to be in the coding region of the transcript where possible, amplicons generally had a length between 500 bp and 2,000 bp. |
| Commercial assay or kit | RNeasy kit | Qiagen | Qiagen: 74106 | |
| Commercial assay or kit | RNase-free DNase | Qiagen | Qiagen: 79254 | |
| Commercial assay or kit | Dynabeads mRNA-purification kit | Invitrogen | Invitrogen: 61006 | |
| Commercial assay or kit | RNA fragmentation kit | Ambion | Ambion: AM8740 | |
| Commercial assay or kit | SuperScript VILO Kit | Invitrogen | Invitrogen: 11754050 | |
| Commercial assay or kit | Mini Quick Spin DNA Columns | Roche | Roche: 11814419001 | |
| Commercial assay or kit | E. coli DNA Polymerase I | Invitrogen | Invitrogen: 18010–025 | |
| Commercial assay or kit | E. coli DNA ligase | Invitrogen | Invitrogen: 18052–019 | |
| Commercial assay or kit | RNase H | Promega | Promega: M4281 | |
| Commercial assay or kit | 5X second strand buffer | Invitrogen | Invitrogen: 10812–014 | |
| Commercial assay or kit | MiniElute Reaction clean-up kit | Qiagen | Qiagen: 28004 | |
| Software, algorithm | Fastqc | https://www.bioinformatics.babraham.ac.uk/projects/fastqc/ | NA | NA |
| Software, algorithm | Cutadapt | https://doi.org/10.14806/ej.17.1.200 | NA | NA |
| Software, algorithm | NextGenMap | http://cibiv.github.io/NextGenMap/ | NA | NA |
| Software, algorithm | Trinity | https://github.com/trinityrnaseq/trinityrnaseq/wiki | NA | NA |

*Continued on next page*

*Continued*

| Reagent type (species) or resource | Designation | Source or reference | Identifiers | Additional information |
| --- | --- | --- | --- | --- |
| Software, algorithm | Cegma | https://doi.org/10.1093/bioinformatics/btm071 | NA | NA |
| Software, algorithm | MUMmer | https://doi.org/10.1002/0471250953.bi1003s00 | NA | NA |
| Software, algorithm | Kraken | https://doi.org/10.1186/gb-2014-15-3-r46 | NA | NA |
| Software, algorithm | bedtools | https://bedtools.readthedocs.io/en/latest/ | NA | NA |
| Software, algorithm | HTSeq | https://doi.org/10.1093/bioinformatics/btu638 | NA | NA |
| Software, algorithm | R | https://www.r-project.org/ | NA | NA |
| Software, algorithm | DESeq2 | Bioconductor | NA | NA |
| Software, algorithm | EdgeR | Bioconductor | NA | NA |
| Software, algorithm | LIMMA | Bioconductor | NA | NA |
| Software, algorithm | ROTS | Bioconductor | NA | NA |
| Software, algorithm | Mfuzz | Bioconductor | NA | NA |
| Software, algorithm | GOstats | Bioconductor | NA | NA |
| Software, algorithm | IQ-TREE | http://www.iqtree.org | NA | NA |

## Animals

*Platynereis dumerilii* were housed in plastic boxes and kept in a 1:1 mixture of Natural Sea water (NSW) and Artificial Sea water (ASW, 30 ‰ Tropic Marine).

Worms were either grown in open culture rooms, or in isolated shelf systems and exposed to a circadian light regime of 16 hr:8 hr light:dark regime (*Figure 1a*), following classical culture conditions (*Hauenschild and Fischer, 1969*), with a circalunar light regime consisting of nocturnal illumination constituting full moon (FM) provided for eight nights, every 21 days (see below for detailed light conditions). To allow for constant availability of mature worms and animals are housed in culture rooms with different circalunar entrainment regimes. Animals from the in-phase (IP) rooms receive the artificial nocturnal light stimulus ('moon') coinciding with the time of full moon in nature; while animals from the out-phase (OP) room receive the moon stimulus when IP room animals are under new moon conditions.

For FRFM sampling, worms were transferred to culture rooms without nocturnal light (i.e. from IP rooms to OP rooms or *vice versa*) prior to the first night of the FM stimulus. Specifics are outlined below and summarised for all samples analysed in *Figure 1—source data 1*.

## Animal staging and head sampling

Animals were sourced from multiple boxes of VIO1 mix and VIO11 mix substrains, originating from the B32134 strain (*Zantke et al., 2014*). Animals were staged based on the visibility of germ cells, and grouped into immature (IM, no germ cells visible), premature (PM), and epitokous non-swarming mature (M) males and females (see *Figure 1b*). Mature animals were defined as having undergone metamorphosis, exhibiting typical yellow (females) or red/white (male) colouration, but were not yet actively swimming (i.e. prior to spawning, *Figure 1b*). Large PM animals were selected for staging 3–5 days prior to sampling.

Animals were first anesthetised for 10 min in a 1:1 mixture of 7.5% (w/v) MgCl$_2$ and ASW. Sex and stage were checked by making a small incision in the trunk wall between two parapodia and gently squeezing out a sample of the coelomic contents. Coelomic cells (eleocytes and oocytes/spermatogonia) were collected in Nereis balanced salt solution (*Heacox et al., 1983*) in a 96-well plate, kept on ice, and imaged using an Axiovert 200M Microscope (Carl Zeiss, Germany), with a Coolsnap

HQ monochrome camera (Photometrics, Vision Systems GmbH, Austria). Staging criteria were devised to obtain male and female animals in equivalent early stages of germ cell maturation. PM males were defined as those with germ cells at late spermatogonia (Spg) I stage as defined by *Meisel, 1990*: characterised by the presence of many large clusters with opaque, smooth, cloud-like appearance, that may also be beginning to dissociate into smaller kidney bean-shaped clusters. PM females were defined as those with a maximum oocyte diameter of 52–75 µm (*Figure 1b*). Oocyte sizes were measured from images taken of oocytes extracted from individual females, using the straight line/measure tool in ImageJ (v1.5g, http://imagej.nih.gov/ij/). Measurements in pixels or inches were converted to µm according to the appropriate scale conversion for the obtained images.

After staging, animals were allowed to recover in NSW/ASW for 24–48 hr. Two days prior to head sampling worms were transferred into sterile filtered NSW containing 0.125 mg/mL ampicilin and 0.500 mg/mL streptomycin-sulphate; after 24 hr, the antibiotics concentration was reduced to 50% for approximately 16 hr, until animals were sampled the following day. For all analyses, pools of 6 heads per biological replicate were sampled at Zeitgeber time 4.

For sampling, animals were anesthetised with $MgCl_2$ as before and the head was cut directly behind the posterior eyes, and in front of the first parapodial segment (red dotted line in *Figure 1b*). Cut heads were quickly rinsed in NBSS and snap frozen in liquid $N_2$ in pools of six in 2 mL tubes containing metal beads. Heads of mature animals were frozen in pools of 2 or three to ensure more efficient RNA extraction.

Samples for initial transcriptome assembly sequencing analysis were obtained from multiple samplings performed in January - March 2014. Biological replicate samples for quantitative transcriptomics were obtained from multiple samplings between November 2014 and February 2015. Samples with best RNA concentrations and quality were selected for sample preparation for RNA-Seq and matched protein fractions were analysed for quantitative proteomics.

## Spectral properties of circadian and circalunar illumination regimes

Light spectra and intensities in the culture room were measured with an ILT950 spectrometer (International Light Technologies Inc Peabody, USA). Data were acquired over 300 ms, and 30 spectra averaged to yield one intensity spectrum, in total five measurement per condition were carried out, and the arithmetic mean of the five spectra was used. Animals grown in an 'in-phase' culture room were exposed to daylight stimulus of 0.0128 $\mu Wm^{-2}$ (3.4566 $\times$ $10^{10}$ photons $s^{-2}m^{-2}$) and 0.0001 $\mu Wm^{-2}$ (4.1480 $\times$ $10^8$ photons $s^{-2}m^{-2}$) full-moon moonlight stimulus; animals grown in an 'out-phase' culture room were exposed to daylight stimulus of 0.0085 $\mu Wm^{-2}$ (2.3090 $\times$ $10^{10}$ photons $s^{-2}m^{-2}$) and 0.0002 $\mu Wm^{-2}$ (6.5192 $\times$ $10^8$ photons $s^{-2}m^{-2}$). Full-moon moonlight stimulus (*Figure 1—figure supplement 2a–d*).

## RNA and protein extraction

Frozen heads were re-suspended in 350 µL RLT-buffer (RNeasy-Kit, Qiagen) and the head capsule was broken for 2 Min at 30 Hz in a Tissue Lyser II (Retsch GmbH). RNA was then extracted following the manufacturer's protocol with additional on-column DNaseI digest. Finally, the RNA was eluted in 30 µL nuclease-free $H_2O$. RNA concentrations and integrity were assessed using the Agilent 2100 Bioanalyzer (Agilent, Nanochip, Total RNA protocol).

To recover proteins from the RNAeasy column flow-through we initially collected all flow-throughs, that is RLT + RW1+RPE. As this combination has a final EtOH content of 45% it readily precipitated proteins when incubated over night at −20°C. To optimise protein recovery we then collected different combinations off column flow-throughs, namely RLT, RLT + RW1, and RLT + RW1 +RPE serving as control. RLT and RLT-RW1 were then precipitated by making them either 50% in acetone, 80% in acetone, or 45% in EtOH to test for the effect of RPE on the precipitation. Analysis of the precipitates by Tricine-SDS-PAGE (*Schägger and von Jagow, 1987*) showed that a four-fold excess of acetone (80% final concentration) added to RLT + RW1 yielded the most quantitative result. Thus, finally the flow-throughs RLT and RW1 were collected and snap frozen in liquid $N_2$ for protein extraction and subsequent LC-MS/MS analysis.

After protein extraction via acetone precipitation (see description in results), we controlled for the presence of non-precipitated proteins in the acetone precipitation supernatant on a monolithic

C18 column and measuring the eluate at 214 nm. For this, the supernatant was dried down in vacuo, reconstituted in 0.1% trifluoro-acetic acid and concentrated on a C18 SPE column from which it was eluted with 90% acetonitrile, 0.1% trifluoro-acetic acid. The UV absorbing compounds eluted in a region typical for peptides and not for proteins.

For the multiplexing strategy of RNA-Seq and proteomic samples see *Figure 1—figure supplement 2e,f*.

## cDNA library preparation for RNA-Seq

The total RNA input used for sample preparation was 700 ng-1,500ng total RNA per sample. PolyA-enriched mRNA was purified from total RNA input using the Dynabeads mRNA-purification kit (Invitrogen, #61006) according to the manufacturer's protocol, and eluted in a final volume of 50 µL. Ribosomal RNA (rRNA) contamination was assessed by Agilent 2100 Bioanalyzer (Total RNA protocol) according to manufacturer's instructions. Samples with rRNA content >25% were subjected to a second round of mRNA purification and re-evaluated by Bioanalyzer as previously. The total volume of purified mRNA was adjusted to 81 µL with nuclease-free water, 9 µL of fragmentation solution (RNA fragmentation kit; Ambion, #AM8740) was added and samples were kept on ice. Fragmentation was carried out at 75°C for 3 min in a thermoblock, at 3 min samples were immediately cooled on ice and 11 µL of stop solution (RNA fragmentation kit, Ambion) was added. Cleanup was performed using the RNeasy Kit and RNA was eluted in 30 µL. Fragment size distribution and concentration was analysed by Bioanalyzer (mRNA protocol).

First strand cDNA synthesis was carried out with the SuperScript VILO Kit (Invitrogen, #11754050) in a 50 µL reaction with an additional 7.5 µg random hexamers (Invitrogen, #48190011) added to the reaction to increase cDNA yield. Reactions were incubated for 10 Min at 25°C (priming), 60 Min at 42°C (reverse transcription), and 5 Min at 85°C (termination). Reaction clean-up was carried out by gel filtration using Sephadex G50-columns (Mini Quick Spin DNA Columns; Roche #11814419001) according to the manufacturer's protocol. Second-strand cDNA was synthesised in a final reaction containing the first strand cDNA reaction and final concentrations of: 550 µM of each dATP, cGTP, dCTP (Thermo Scientific, #R0181), and dUTP (deoxy-UTP sodium salt; Roche, #11934554001); 0.300 U DNA Polymerase I (*E.coli*; Invitrogen, #18010–025), 0.073 U *E. coli* DNA ligase (Invitrogen, #18052–019), and 0.015 U RNase H (Promega, #M4281); 20 µL 5X second strand buffer (Invitrogen, #10812–014), in a final volume of 100 µL. Reactions were incubated for 2 hr at 16°C then purified with by the MiniElute Reaction clean-up kit (Qiagen, #28004), eluted in 2 × 10 µL volumes in a single reaction tube. Samples were stored at −20°C prior to final library preparation for sequencing.

## RNA-Seq library preparation and sequencing

Quality control for concentration, size selection, adapter ligation, pre-amplification PCR and dilution for sequencing was performed according to standard procedures and carried out by the VBCF Next-Generation Sequencing facility services (VBCF-NGS, Vienna Biocenter Campus, https://www.vbcf.ac.at/facilities/next-generation-sequencing/). Samples for initial transcriptome assembly were sequenced on the HiSeq2000 (Illumina) in single lane format (one sample per lane for each of: IM-NM +FM combined, IM-FRFM, PM_NM_male, PM_NM_female, PM_FRFM_male, PM_FRFM_female, M_NM_male, M_NM_female). The IM_NM + FM sample was run in paired-end 100 bp format (PE100), all other samples were analysed as single-end 100 bp (SE100). Biological replicate samples for differential expression analysis were run using SE100 sequencing over seven lanes, in a 6 × 5 plex (*Figure 1—figure supplement 2e*) and 1 × 6 plex format on the HiSeq2500 instrument (Illumina). The 6-plex lane was a technical replicate lane and contained a single technical replicate from IM samples of each of the other six 5-plex lanes. Read data was quality checked, de-multiplexed and converted to. bam format by NGS facility bioinformatics staff (VBCF-NGS). Reads for individual sequencing libraries are deposited in the European Nucleotide Archive (ENA) under accession number PRJEB27496 (*Figure 2—source data 4*).

## Transcriptome assembly, curation, annotation and read mapping

Sequence reads from the eight libraries sequenced on single lanes for deep coverage were used for transcriptome assembly. Paired-end reads from the IM_NM + FM sample were used as a scaffold for the base and combined with the SE100 reads from the other seven libraries for the final assembly.

The raw Illumina reads (totalling 1,667,584,873 reads) were quality checked using Fastqc. Cutadapt (v1.9.1, *Martin, 2011*) to remove the adapter sequences (-b option) of the paired end reads and to quality trim the reads (-q 20) requiring that the remaining sequence was at least 13 bp long (-m 13). We used ngm-utils interleave (v0.4.5) to reorder the fastq files and filter out reads not having a paired read anymore. The so processed reads were merged and used as input for the RNA-Seq assembly with Trinity (v2.0.2, *Grabherr et al., 2011*). Paired end as well as singleton reads were provided as input to Trinity which was run without taking the strand into account. This initial Pdu_HeadRef_TS_v1 assembly comprised a total of 407,172. Of these multiple copies, only the longest sequence per cluster was selected and this Pdu_HeadRef_TS_v2 assembly was filtered by size to retain transcripts greater than 500 bp in size leading to Pdu_HeadRef_TS_v3 with 64,335 sequences. Finally, the completeness of our RNA-Seq assembly was tested using Cegma (v2.5, *Parra et al., 2007*), revealing that 85.1% of the genes predicted to be present were present full length, and 93.6% were present either full length or partially.

The Pdu_HeadRef_TS_v3 assembly was then analysed for eukaryotic contaminations using MUMmer (v3.23, *Delcher et al., 2003*) with a customised script (https://github.com/fritzsedlazeck/sge_mummer) to check for exact hits to common model organism genomes [human (GRCh38), mouse (GRCm38), zebrafish (Zv9), medaka (MEDAKA1), *Arabidopsis* (TAIR10), *Drosophila* (BDGP6), *C. elegans* (WBcel235)], as well as dinoflagellates (*Oxyrrhis*) and algae (*Tetraselmis*) used as *Platynereis* food. *Platynereis* sequences which had 90% or higher identity and at least 90% of the RNA-Seq scaffold aligned with sequences from the above mentioned organisms were considered putative contaminations. In order to prevent the exclusion of highly conserved *Platynereis* orthologues, these identified putative eukaryotic contaminations were not excluded from the list, but marked with a tentative suffix ('_contamination') to enable them to be distinguished, investigated and post-filtered from downstream analyses, if necessary.

Further curation was then applied to remove redundancy; and these sequences were assembled together with a previously published *Platynereis dumerilii* transcriptome (herein refered to as Conz_TS: *Conzelmann et al., 2013*) containing ~350,000 individual contigs. In addition we checked for the presence of all *Platynereis* transcripts present in GenBank and added/replaced shorter assembled contigs from our assembly with the NCBI sequences where necessary. These sequences were aligned with MUMmer and replaced the scaffolds in our transcriptome if the alignment showed at least 90% identity, at least 200 bp and more than 70% of the scaffold were aligned to the previously published scaffold and the published sequences was longer then our scaffold (if the transcript was absent the sequence from NCBI was added; if the contig from our assembly was shorter than the ncbi sequence then the contig was replaced by the ncbi sequence). We then self-aligned the so obtained scaffolds using MUMmer and kept only one copy of the sequence if the sequences had over 98% identity and one of the copies is covered at least 99% of the length of the other and was longer, otherwise both sequences were kept. This resulted in Pdu_HeadRef_TS_v4 containing 57,869 total sequences. For this assembly we obtained the longest open reading frame (ORF) for all six frames (Pdu_HeadRef_prot_v4) by using Transdecoder (v2.0.1, http://transdecoder.sourceforge.net) and keeping the longest ORF per frame.

To identify prokaryotic contaminations we used Kraken (v0.10.6, *Wood and Salzberg, 2014*), which uses k-mer matching to identify hits between the query sequences (Pdu_HeadRef_TS_v4) with the default database. To avoid falsely marking orthologues genes we only marked scaffolds that aligned to two different genomes. This approach identified 6,227 putative contigs of prokaryotic origin. We identified 500 of these to be verified *Platynereis* genes from GenBank, and thus constituting false positives, leaving a total of 5,727 contaminating sequences (9.9% of the total TS_v4 transcriptome, *Figure 2—source data 1*). These, together with 83 transcripts spiking in one sample (IM_NM) discovered during data processing for differential expression analysis (see below) were then removed to generate the final Pdu_HeadRef_TS_v5 assembly containing 52,059 transcripts.

Base annotation of the TS_v1 assembly was performed using Trinnotate (v2.02) to identify blastx, blastp, HMMER, GO terms etc. for each contig (*Figure 2—source data 2*). The Pdu_HeadRef_TS_v5 assembly was aligned using blastx against the NCBI_nr_protein database (v_April_2016) and NCBI_ensembl_mouse protein database (v_April_2016) to identify putative protein orthologues with e-value cut-offs at $10^{-4}$.

Protein domains, GO term annotations, PANTHER IDs, and pathway analysis etc was additionally analysed and reported for all 57,869 transcripts using InterProScan (v5, using standard parameters).

Nucleotide sequences were used as input, split into 18 subsets of ≤3,000 transcripts and first translated using EMBOSS getorf (http://emboss.sourceforge.net/apps/cvs/emboss/apps/getorf.html) and translated sequences were then subject to InterProScan analysis. The results were compiled into a single file merged in addition with the protein_nr and mouse_nr blastx results (*Figure 2—source data 3*). Further individual reciprocal blastx searches against ncbi_nr protein database were performed in addition to phylogenetic analyses to confirm the orthology of specific candidate genes of interest.

## Read mapping and differential expression analysis of RNA-Seq data and selection of top regulated candidates for validation

The curated Pdu_HeadRef_TS_v4 assembly was used to map all sequencing reads (adapter filtered and quality trimmed) from the replicate head samples for differential expression analysis using Next-GenMap (v0.4.13, *Sedlazeck et al., 2013*) with default parameters. Subsequently, reads were quality filtered (-q20) and converted to a bam file using samtools. The coverage per scaffold was assessed using bedtools (v2.17.0) multicov. The obtained Pdu_HeadRef_TS_v4 count data (generated by HTseq v0.6.1) were processed by first, collapsing the technical replicates (see above and *Figure 1—figure supplement 1e*) by sum, then by filtering out 83 transcripts with extremely uneven counts in IM_NM sample [counts IM_NM.2>=10 x counts(IM_NM.1+IM_NM.3)] and finally by removing the 5,727 contaminations identified by KRAKEN leading to a 52,059 transcripts (equivalent to Pdu_HeadRef_TS_v5) containing input table (S5_Pdu_HeadRef_TS_v5_ReadCounts). Then differential expression analysis was performed for 14 pairwise comparisons (*Figure 1—figure supplement 1*, *Figure 1—source data 2*) and the DESeq derived normalised read counts were used to plot gene expression profiles (*Figure 3—source data 2*) using the R packages DESeq2 (v1.10.0, *Love et al., 2014*) and EdgeR (v3.12.0, *Robinson et al., 2010*). DESeq2 was run with standard parameters and EdgeR was run applying the generalised linear model (*Hughes et al., 2012*) algorithm. Output lists were cut off using a Benjamini-Hochberg FDR (*Benjamini and Hochberg, 1995*) of 0.1 (10%).

To filter candidate lists for validation, separate lists of differentially expressed transcripts (DETs) from DESeq2 and EdgeR were merged by transcript ID and rankings were applied separately to the DESeq2 and EdgeRGLM-derived lists, where transcripts were assigned a rank from one to *n* (where *n* is the number of transcripts), based on ranking by lowest to highest adjusted p-value (padj/FDR). The DESeq2 and EdgeRGLM rank values for each transcript were then added to give a rank sum value, and the table was re-sorted by this rank sum value from lowest to highest. For maturation comparisons, 'unisex' candidate lists were further defined as subsets of transcripts predicted as differentially-expressed between stages, for both male and female comparisons (e.g. if transcript A is up-regulated in IM_NM vs PM_NM_male and up-regulated IM_NM vs PM_NM_female, it is classed as 'unisex' up-regulated for IM_NM vs PM_NM). A schematic overview of the comparison procedure can be found in *Figure 1—figure supplement 1*, and see also *Figure 1—source data 2*.

## Mass spectrometry

Proteins for MS analysis were extracted from the same samples used for RNA sequencing, by collecting the first two flow-through fractions from the SiO$_2$-based RNA extraction columns. The pooled flow-through samples were snap frozen in liquid nitrogen and stored at −80°C until further use.

To recover proteins, the flow-through fractions were first lyophilised then re-suspended in H$_2$O to a final volume of 1.0 mL and 4.0 mL of cold acetone was added and proteins were precipitated over-night at −20°C. The precipitate was then collected by centrifugation for 30 min by centrifugation at 3,500xg at 4°C. The resulting pellet and a quarter of the supernatant were transferred in to a fresh tube and the remaining supernatant split into three equal parts which were then centrifuged for 30 Min at 14,000xg and 4°C to collect the remaining proteins. Pellets were then washed with 100 µL −20°C-cold EtOH to remove the co-precipitated GuSCN from the RNA-extraction buffer and allowed to dry over-night, dissolved in 100 µL UA-buffer [8M urea, 50 mM triethylammonium bicarbonate (TEAB), pH 8.5] by sonication, and finally re-pooled. Proteins were digested for MS/MS-analysis using the Filter Aided Sample Preparation method (*Wiśniewski et al., 2009*). Briefly, the samples were reduced by 10 mM DTT for 30 Min at RT in UA, applied to an YM-30 filter membrane, centrifuged and washed. Thiols were alkylated for 30 Min with 50 mM iodoacetamide at RT in UA centrifuged and washed with 50 mM TEAB. Proteins were then digested with trypsin (1:50 trypsin:

protein ratio) overnight at 37°C in 50 mM TEAB. Digested peptides were eluted from filters by centrifugation and acidified by the addition of trifluoroacetic acid, and desalted by solid phase extraction (SPE) with Sep-Pak C18 columns (Waters). Eluates were dried in a speed-vac and the resulting pellets were dissolved in 100 μM TEAB and labelled with TMT isobaric tags (TMTsixplex Isobaric Label Reagent, Thermo Fisher Scientific) according to manufacturer's protocol. Labelling efficiency was assessed by a LC-MS/MS run on Q Exactive plus Orbitrap mass spectrometer (Thermo Fisher Scientific) for each label separately and found to be complete when semi- or unlabelled peptides were less than 1% of total PSMs. Once maximal labelling efficiency was confirmed the reaction was quenched with 8 μL 5% hydroxylamine per reaction vial (15 min incubation time at room temperature). FASP-digested and labelled samples were then combined in a 5-plex multiplex format for Set1 and a 6-plex format for Set2. Peptides were separated on an Ultimate 3000 RSLC nano-flow chromatography system using a pre-column for sample loading (PepMapAcclaim C18, 2 cm ×0.1 mm, 5 μm,) and a C18 analytical column (PepMapAcclaim C18, 50 cm ×0.75 mm, 2 μm, all Dionex-Thermo Fisher Scientific), applying a linear gradient over for 4 hr from 2% to 35% solvent B (80% acetonitrile, 0.1% formic acid; solvent A 0.1% formic acid) at a flow rate of 230 nL/min. Eluting peptides were analysed on a Q Exactive Plus Orbitrap mass spectrometer equipped with a Proxeon nanospray source (all Thermo Fisher Scientific), operated in a data-dependent mode. Survey scans were obtained in a mass range of 380–1,650 m/z with lock mass off, at a resolution of 70,000 at 200 m/z and an AGC target value of $3 \times 10^6$. The 12 most intense ions were selected with an isolation window width of 1.2 Da, fragmented in the HCD cell at 35% collision energy and the spectra recorded at a target value of $1 \times 10^5$ and a resolution of 17,500. The peptide match and exclude isotope features were enabled and selected precursors were dynamically excluded from repeated sampling for 40 s. Five technical replicate runs were analysed for each set of multiplexed samples.

Data analysis was performed with Proteome Discoverer 2.1 using Sequest to search against the *Platynereis* proteome database (Pdu_HeadRef_prot_v4, ProteomeXchange: identifier PXD010532). Carbamidomethylation of cysteines, TMT-labelling on peptide N-termini and lysines were selected as fixed modifications, oxidation of methionine as variable modification. Precursor tolerance was set to 10ppm, fragment tolerance to 0.02 Da. Data were filtered for an FDR of 1% at PSM and protein level. Quantification was performed via the reporter ions quantifier mode in PD 2.1 based on the signal/noise values (herein referred to as intensity). Spectra with isolation interference greater 30% were not used for quantification. Protein intensity values were calculated for each biological replicate as the average intensity of all peptides per protein across all five technical replicates (herein referred to as Mean Intensities per protein). Mean intensities per protein were then normalised between isobaric ion channels to obtain a 1:1:1:1:1 ratio of reporter ion intensities based on the summed reporter intensities across all proteins per channel, within each set of 5 multiplexed samples (herein these expression values are referred to as Norm. Mean intensities per protein). Quantifiable proteins were further defined as those with at least two uniquely mapping peptides and detected in at least 2/3 biological replicates per sample group.

To normalise for variation in protein intensity distributions between individual samples, an additional channel median normalisation was applied, whereby the median protein intensity value was calculated for each sample ('channel median value') and all individual protein values per sample were then divided by this channel median value. We performed an additional inter-replicate median normalisation to account for inter-run variation between biological replicate sets within each of the NM and FRFM experiments, by: i) calculating for each biological replicate set; the median intensity across all five samples per protein, ii) calculating conversion factors to equalize the median intensity per protein between biological replicate sets. Conversion factors (CF) were calculated for the NM set relative to BR1, for example: CF.BR2 = Median_NM_BR1/Median_NM_BR2; CF.BR3 = Median_NM_BR1/Median_NM_BR3, and relative to BR3 for the FRFM set, for example: CF.BR1 = Median_FRFM_BR3/Median_FRFM_BR1; CF.BR2 = Median_FRFM_BR3/Median_FRFM_BR3. All intensity values per protein per biological replicate were then multiplied by the respective conversion factor. To enable normalisation and comparison of the data from the NM and FRFM experiments we included a spiked-control sample from the NM_BR1 set (TMT-128_PM_NM_female.1) in each of the FRFM biological replicate sets. Therefore, as a final step the FRFM intensity values were normalised relative to those of the NM_BR1 set by multiplying the values for each protein in each BR set, by corresponding conversion factors (CF) calculated as the ratio of the NM_BR1 TMT-128_PM_NM_female1 intensity/intensity value recorded for this sample in FRFM BR sets 1, 2 and 3

(e.g. CF.FRFM1 = TMT-128_PM_NM_female1:NM/TMT-128_PM_NM_female1:FRFM_BR1; CF. FRFM2 = TMT-128_PM_NM_female1:NM/TMT-128_PM_NM_female1:FRFM_BR2). The effect of normalisation resulted in median-centred data and decreased variability in the distribution of protein intensity values between biological replicates (*Figure 5—figure supplement 7* and *Figure 5— source data 1*). TMT-based quantitative proteomics data were acquired using the VBCF instrument pool (www.vbcf.ac.at).

For benchmarking the protein set obtained with the described method, results were also compared with a set of proteins identified by a more conventional unlabelled proteomic analysis on separate head samples (Bileck, Gerner et al., unpublished). Briefly, for this analysis, tissue samples were incubated in 100 µl sample buffer (7.5M urea, 1.5M thiourea, 4% CHAPS, 0.05% SDS, 100 mM DTT) for 4 hr and lysed using an ultrasonic probe. An adapted filter aided sample preparation protocol was used for protein digestion as described (*Slany et al., 2016*). Cleaned peptide samples were reconstituted in 5 µl 30% formic acid containing 10 fmol each of 4 synthetic standard peptides and diluted with 40 µl mobile phase A (98% $H_2O$, 2% acteonitrile, and 0.1% formic acid) and analysed as described previously (*Slany et al., 2016*). Identification of proteins was performed using the Max-Quant software (v1.5.2.8) including the Andromeda search (*Cox and Mann, 2008*) to map the identified peptides against the Pdu_HeadRef_prot_v4 database. A minimum of two peptide identifications, at least one of them unique, was required for positive protein identification. For peptides and proteins, a FDR of less than 0.01 was applied.

## Differential abundance analysis of proteomics data

Differential abundance analysis for the normalised proteomic data for the subset of 1064 quantifiable proteins was performed using both Linear Models for Microarray and RNA-Seq Data (*Li et al., 2016*, http://bioinf.wehi.edu.au/limma) and the Reproducibility-Optimised Test Statistic (ROTS) package for R (*Elo et al., 2008*; *Elo et al., 2009*, http://www.btk.fi/research/research-groups/elo/software/rots/). Both ROTS and LIMMA are able to handle missing values, which is a common feature of proteomic datasets. ROTS and LIMMA were used to analyse differential expression using a BH FDR of 0.1 (10%), for the 14 pairwise comparisons (*Figure 1—figure supplement 1*, *Figure 1— source data 2*) defined previously for RNA-Seq differential expression analysis using DESeq2 and EdgeR, details with regard to the statistical testing are given below in the section Statistics. The same rank sum approach was taken to merge the results of LIMMA and ROTS predictions and rank candidates according to adjusted pvalues. Candidates were further classified into unisex groupings for maturation stage comparisons.

## Testing for statistical effects in the number of analysed transcripts

To test whether the different input list sizes of transcripts and proteins, 52,059 and 1,064 respectively, might affect the relative amount of DE predictions, we also performed DESeq2 and EdgeR analysis on the 1000 and 5,000 most abundant transcripts. For this, we calculated the mean read counts over all comparison per transcript, sorted the transcript list and chose the 1,000 or 5,000 transcripts with the highest average number of read counts. These reduced lists were then subjected to DE analysis by DESeq2 and EdgeR as before, and rank sum lists of these analyses were then generated as outlined above.

## Quantitative PCR

qRT-PCR, including primer design, primer testing for efficiency, cDNA synthesis protocol, reaction chemistry, cycle conditions and instrumentation, was performed as described previously (*Schenk et al., 2016*), with the following modifications: Total RNA (200 ng per sample) was reverse transcribed to cDNA using the Quantitect reverse transcription kit (Qiagen, Cat# 205310), with incubation at 42°C for 20 min for reverse transcription. Final cDNA reactions were diluted to a final volume of either 95 µL for low-expression level and lunar candidate qRT-PCR assays, or 350 µL for sexbiased and maturation candidate qPCR assays and all qRT-PCR reactions were performed in 15 µl final volume, using 3 µL of cDNA template. Primers for qPCR assays (intron-spanning where possible) were designed using the Universal Probe Library assay design tool (Roche, *Supplementary file 2* ).

Target genes and reference controls were analysed in duplicate reactions for all samples. Plate control cDNA and –RT controls for the target gene(s) were included on each plate as quality controls

for inter-plate consistency. *Cell cycle serine/threonine kinase* (*cdc5*, *Dray et al., 2010*; *Zantke et al., 2013*) and *S-adenosyl methionine synthetase* (*sams*, *Schenk et al., 2016*), were measured as reference genes, based on previous evaluation as stable qPCR reference controls (*Schenk et al., 2016*). Expression levels for target genes were calculated using the $\Delta$-$C_t$ method, using the mean of *cdc5* and *sams* $C_t$ values for relative normalisation (*Figure 4—source data 1*). Calculated $\Delta$-$C_t$ values were exponentially transformed ($2^{-(\Delta-C_t)}$) to give the final normalised relative expression values. Statistical analysis was performed as outlined below.

## Sequence analysis tools

CLC Main workbench (version seven and higher) were used for routine sequence analysis, small-scale BLASTx and BLASTp searches against NCBI, assemblies, basic multiple sequence alignments, and sequence annotations. Phylogenetic trees for ependymin-related proteins (ERPs) and Whitnin/Proctolin were generated from multiple sequence alignments (MUSCLE: http://www.drive5.com/muscle, *Edgar, 2004*), see *Figure 6—source datas 1* and *2*; *Figure 7—source datas 1* and *2*) and IQ-TREE (v1.6.3, *Minh et al., 2013*; *Nguyen et al., 2015*), ultra fast bootstrapping (*Hoang et al., 2018*), and the integrated ModelFinder (*Kalyaanamoorthy et al., 2017*).

Simple Modular Architecture Research Tool (SMART, http://smart.embl-heidelberg.de) was used for analysis of protein domains in individual sequences. Functional descriptions of individual orthologue proteins were also sourced from the Uniprot database (http://www.uniprot.org).

## Cloning and plasmids used in this study

Candidate genes of interest were cloned from cDNA reverse transcribed from total RNA of heads from appropriate sex, stage and circalunar condition, using the Transcriptor high fidelity cDNA synthesis kit (Roche, #05081955001). Final cDNA reactions were diluted to a final volume of 50 µL with nuclease free water. PCR: 20 µl final reaction volume, 1–5 µl of cDNA template, Phusion polymerase (Thermo Scientific). PCR programs: 98°C denaturing cycles of 30 s, annealing temperatures ranging from 58–62°C with 20 s annealing times and extension at 72°C for 35–40 cycles (primer details- *Supplementary file 2*). PCR products were analysed by agarose gel electrophoresis (GelGreen, Biotium), bands of expected size were isolated using the Qiagen gel extraction kit according to the manufacturer's instructions. Extracted DNA was ligated into the pJET1.2 vector DNA (CloneJET PCR cloning kit, Thermo Scientific). Plasmid DNA was purified after colony PCR using a standard alkaline lysis protocol with sequential precipitation using isopropanol and 70% ethanol (*Birnboim and Doly, 1979*). DNA pellets were resuspended in 30–50 µL of nuclease free water and verified by Sanger sequencing (Microsynth, Austria). Plasmid cloning info- *Supplementary file 2*.

## Whole mount in situ hybridisation

Animals for in situ hybridisation (ISH) were sourced from culture boxes, sexed and staged according to the aforementioned criteria and maintained in separate boxes in antibiotic-free ASW under respective circalunar conditions for at least 48 hr prior to sampling. ISH for head tissues was carried out as previously described (*Tessmar-Raible et al., 2005*; *Zantke et al., 2013*). DNA templates for in vitro transcription were generated by amplifying sense and antisense sequences from vector DNA templates using Sp6-conjugated pJET1.2 forward and reverse primer combinations (see *Supplementary file 2*). Probes were purified using the RNeasy kit (Qiagen) according to manufacturer's instructions, eluted in 30 µL of RNase-free water, probe quality analysed by agarose gel electrophoresis and diluted to a concentration of 100 ng/µL or 50 ng/µL with hybridisation solution and stored at −20°C. For ISH 500 or 1000 ng of probe were used depending on the expression of the gene as judged from RNA-Seq. Stained tissues were stored in 80–100% glycerol/1X PTW at 4°C in 2 mL tubes.

For all genes assessed sense controls were run in parallel in pools of 3–5.

*In situs* were imaged using an Axioplan Z2 Microscope (Carl Zeiss, Germany), with AxioCam MRc5 colour CCD camera (Carl Zeiss, Germany) and captured using ZenPro Software (v2.0, Carl Zeiss, Germany). Images were saved in TIFF format.

## Image analysis and processing

Where necessary brightness and contrast adjustments and cropping of images in TIFF format was carried out using ImageJ (v1.50i) or Adobe Photoshop CC (v2015). Figures were constructed using Adobe Illustrator CC (v2015).

## Data analysis methods RNA-Seq and proteomics functional analyses

### Soft clustering for identification of similarly regulated genes

Expression profile clustering was performed using the Mfuzz package (v2.32.0, *Futschik and Carlisle, 2005*; *Kumar and Futschik, 2007*, http://mfuzz.sysbiolab.eu) for R. Briefly, input tables of mean read counts, or mean normalised protein intensity values were organised in columns with separate rows for each protein or transcript. Input data was standardised using the standardise() function of the Mfuzz package and the estimate of fuzziness (m1) was calculated using mestimate(). The number of clusters was determined by analysing cluster dendograms generated from a distance matrix using hclust() and cutree() to visualise different cluster numbers relative to the dendogram branching. Cluster information was used to group candidate transcript and protein expression profiles by likeness and membership were used as a measure of fit for individual profiles to the cluster mean profile.

### GOstats analysis for predictive functional annotation

Gene ontology (GO) analysis was performed using the GOstats package (v2.46.9, *Falcon and Gentleman, 2007*), adapting the procedure described by Carlson for uncommon model organisms (https://www.bioconductor.org/packages/release/bioc/vignettes/GOstats/inst/doc/GOstatsForUnsupportedOrganisms.pdf). Briefly, the background gene set (gene universe) files (*Figure 3—source data 3*) were generated using GO terms retrieved from the InterProScan results containing evidence code 'IAE'. For GO enrichment testing of differentially expressed genes (DEGs) from RNA-Seq experiments, and for testing over and under-represented GO ids in the detected set of proteins against the transcriptome, the gene universe was derived from the full set of GO terms for the whole Pdu_HeadRef_Ts_v5 transcriptome, and comprised 2,472 unique GO terms corresponding to 16,921 transcripts (*Figure 3—source data 3*). For differential expression of proteins the gene universe size was derived from the set of 2,290 quantifiable proteins, and comprised 1,035 GO terms corresponding to 1,751 proteins (*Figure 5—source data 2*). To perform hypergeometric testing using the selected gene universe, GO2All mapping was first performed using the Human AnnotationDbi package and the resulting GOALLFrame object was used to generate a customised GeneSetCollection for the hypergeometric testing, leading to 4,775 unique GO-terms in the transcriptome universe; and 2,448 unique GO-terms in the protein background. Tests for over and under-representation of GO terms were run for all sets of candidates, for all ontologies (Biological Process: BP, Molecular Function: MF, Cellular Compartment: CC) using conditional parameters. Ranking of GO enrichment was performed using the p-value, which indicates the relative significance of enrichment of genes in the tested gene list compared to the specified gene universe.

Individual ontologies (BP, MF, CC) were analysed for over and/or under-representation for each subset of candidate transcripts/proteins and enriched GO terms were linked to corresponding transcript ids using the original gene universe file, and by mapping to parent terms using acyclic graphs from the GO.db package.

## Statistics

All statistical analyses were performed with R (*R Core Development Team, 2015*), and various statistical packages available for R.

Analysis of differentially expressed genes was performed with EdgeR (*McCarthy et al., 2012*; *Robinson et al., 2010*) and DESeq2 (*Love et al., 2014*) using custom made scripts, considering a BH FDR of 0.1 as statistically significant.

Proteomic data were analysed using either LIMMA (*Ritchie et al., 2015*) and ROTS (*Elo et al., 2008*; *Elo et al., 2009*). LIMMA uses linear modelling to test the fit of obtained expression data to the null hypothesis based on the defined experimental design. The script used was modified based on a previously available application for labelled-proteomics data (*Kammers et al., 2015*, http://www.biostat.jhsph.edu/~kkammers/software/eupa/R_guide.html), and used the more conservative

BH FDR rather than q-values (*Storey and Tibshirani, 2003*) originally in the script. ROTS is a modified form of the t-test, which uses bootstrapping to analyse differential expression between sample groups using random-subsets of samples. From the collective results a modified test statistic is calculated based on the reproducibility of the list of differentially expressed predictions (*Elo et al., 2008*).

GO-Term analysis was performed using the GOstats package (v2.46.0), which applies a hypergeometric test to test for significance. We used the conditional algorithm of hypergeometric test which takes into account the hierarchical structure of the GO-terms by testing the leaves (GO-Terms with no child-terms) first. The conditioning then leads to the elimination of those genes associated with a GO-Term which have been tested as significant in the GO-term's children (*Falcon and Gentleman, 2007*).

qRT-PCR data were first checked for normality by the Shapiro-Wilk test (*Shapiro and Wilk, 1965*) using the generic shapiro.test() function. Statistical significance was then assessed by an unpaired t-test with Welch's correction for heteroskedasticity (*Welch, 1947*) with at least five biological replicates per group. Corrections for multiple testing were done applying the BH algorithm (*Benjamini and Hochberg, 1995*).

## Acknowledgements

We thank the members of the Tessmar-Raible and Raible groups for discussions, Andreij Belokurov and Margaryta Borysova for excellent worm care at the MFPL Fish and Marine facility.

## Additional information

### Competing interests

Stephanie C Bannister: is affiliated with Lexogen GmbH. The author has no other competing interests to declare. The other authors declare that no competing interests exist.

### Funding

| Funder | Grant reference number | Author |
| --- | --- | --- |
| European Research Council | (FP7/2007-2013)/ERC Grant Agreement 260304 | Florian Raible |
| European Research Council | (FP7/2007-2013)/ERC Grant Agreement 337011 | Kristin Tessmar-Raible |
| University of Vienna | Research Platform Rhythms of Life | Arndt von Haeseler Christopher Gerner Florian Raible Kristin Tessmar-Raible |
| Austrian Science Fund | #AY0041321 | Kristin Tessmar-Raible |
| Austrian Science Fund | #P28970 | Kristin Tessmar-Raible |
| Austrian Science Fund | #I2972 | Florian Raible |
| Austrian Science Fund | #M1478 | Sven Schenk |
| Vienna International PostDoctoral Program for Molecular Life Sciences | #GM100402 | Stephanie C Bannister |
| European Molecular Biology Organization | YIP #2893 | Kristin Tessmar-Raible |

The funders had no role in study design, data collection and interpretation, or the decision to submit the work for publication.

### Author contributions

Sven Schenk, Stephanie C Bannister, Conceptualization, Data curation, Formal analysis, Investigation, Methodology, Writing—original draft, Writing—review and editing; Fritz J Sedlazeck, Bui Quang Minh, Data curation, Software, Writing—review and editing; Dorothea Anrather, Formal

analysis, Methodology, Writing—review and editing; Andrea Bileck, Data curation, Methodology, Writing—review and editing; Markus Hartl, Kristin Tessmar-Raible, Conceptualization, Resources, Supervision, Funding acquisition, Investigation, Methodology, Writing—original draft, Project administration, Writing—review and editing; Arndt von Haeseler, Resources, Supervision, Writing—review and editing; Christopher Gerner, Resources, Methodology, Writing—review and editing; Florian Raible, Conceptualization, Resources, Supervision, Funding acquisition, Validation, Investigation, Writing—original draft, Project administration, Writing—review and editing

### Author ORCIDs
Sven Schenk http://orcid.org/0000-0002-7689-5854
Fritz J Sedlazeck https://orcid.org/0000-0001-6040-2691
Bui Quang Minh http://orcid.org/0000-0002-5535-6560
Andrea Bileck http://orcid.org/0000-0002-7053-8856
Markus Hartl https://orcid.org/0000-0002-4970-7336
Arndt von Haeseler http://orcid.org/0000-0002-3366-4458
Florian Raible http://orcid.org/0000-0002-4515-6485
Kristin Tessmar-Raible http://orcid.org/0000-0002-8038-1741

### Decision letter and Author response
Decision letter https://doi.org/10.7554/eLife.41556.113
Author response https://doi.org/10.7554/eLife.41556.114

## Additional files
### Supplementary files
• Supplementary file 1. Table containing the IDs of the 3,847 identified proteins using an unlabelled proteomics approach.
DOI: https://doi.org/10.7554/eLife.41556.080

• Supplementary file 2. Table showing sequences and melting temperatures for the primers used for qPCR and cloning in ths study, as well as information on the plasmids generated.
DOI: https://doi.org/10.7554/eLife.41556.081

• Source data 1. Short description of all source files and supplementary files.
DOI: https://doi.org/10.7554/eLife.41556.082

• Transparent reporting form
DOI: https://doi.org/10.7554/eLife.41556.083

### Data availability
The data generated and analysed during this study are included in the published article, its supplementary information files, and the following repositories: The mass spectrometry proteomics raw data and the translated protein database FASTA file have been deposited in the PRIDE database repository (Vizcaíno et al., 2016) with the dataset identifier PXD010532. Transcriptome data (RNA-Seq read libraries) have been submitted to the sequence read archive (SRA) through ENA; accession numbers: see Figure 2—source data 4. Transcriptome data (assembled sequences): ENA repository; accession number: PRJEB27496. Sequences of cloned coding sequences validated in this study: GenBank repository; accession numbers: MH587646-MH587650, MH645920-MH645924, MH678618 and MH678619.

The following datasets were generated:

| Author(s) | Year | Dataset title | Dataset URL | Database and Identifier |
|---|---|---|---|---|
| Schenk S, Stephanie C Bannister, Fritz J Sedlazeck, Dorothea Anrather | 2019 | Combined transcriptome and proteome profiling reveals specific molecular brain signatures for sex, maturation and circalunar clock phase | https://www.ncbi.nlm.nih.gov/nuccore/MH645920 | NCBI GenBank, MH645920 |
| Schenk S, Bannister | 2019 | Combined transcriptome and | https://www.ncbi.nlm. | NCBI GenBank, |

| | | | | |
|---|---|---|---|---|
| S, Fritz J Sedlazeck, Dorothea Anrather | | proteome profiling reveals specific molecular brain signatures for sex, maturation and circalunar clock phase | nih.gov/nuccore/ MH678618 | MH678618 |
| Schenk S, Bannister S, Sedlazeck FJ, Anrather D, Minh BQ | 2019 | Combined transcriptome and proteome profiling reveals specific molecular brain signatures for sex, maturation and circalunar clock phase | https://www.ebi.ac.uk/ ena/data/view/ PRJEB27496 | European Nucleotide Archive, PRJEB27496 |
| Schenk S, Bannister S, Sedlazeck FJ, Anrather D, Minh BQ, Bileck A, Hartl M, von Haeseler A, Gerner C, Raible F, Tessmar-Raible K | 2019 | Combined transcriptome and proteome profiling reveals specific molecular brain signatures for sex, maturation and circalunar clock phase | https://www.ebi.ac.uk/ pride/archive/projects/ PXD010532 | EBI PRIDE, PXD010 532 |
| Schenk S, Bannister S, Sedlazeck FJ, Anrather D | 2019 | Combined transcriptome and proteome profiling reveals specific molecular brain signatures for sex, maturation and circalunar clock phase | https://www.ncbi.nlm. nih.gov/nuccore/ MH587646 | NCBI GenBank, MH587646 |
| Schenk S, Bannister S, Sedlazeck FJ, Anrather D | 2019 | Combined transcriptome and proteome profiling reveals specific molecular brain signatures for sex, maturation and circalunar clock phase | https://www.ncbi.nlm. nih.gov/nuccore/ MH587647 | NCBI GenBank, MH587647 |
| Schenk S, Bannister S, Sedlazeck FJ, Anrather D | 2019 | Combined transcriptome and proteome profiling reveals specific molecular brain signatures for sex, maturation and circalunar clock phase | https://www.ncbi.nlm. nih.gov/nuccore/ MH587648 | NCBI GenBank, MH587648 |
| Schenk S, Bannister S, Sedlazeck FJ, Anrather D | 2019 | Combined transcriptome and proteome profiling reveals specific molecular brain signatures for sex, maturation and circalunar clock phase | https://www.ncbi.nlm. nih.gov/nuccore/ MH587649 | NCBI GenBank, MH587649 |
| Schenk S, Bannister S, Sedlazeck FJ, Anrather D | 2019 | Combined transcriptome and proteome profiling reveals specific molecular brain signatures for sex, maturation and circalunar clock phase | https://www.ncbi.nlm. nih.gov/nuccore/ MH587650 | NCBI GenBank, MH587650 |
| Schenk S, Bannister S, Sedlazeck FJ, Anrather D | 2019 | Combined transcriptome and proteome profiling reveals specific molecular brain signatures for sex, maturation and circalunar clock phase | https://www.ncbi.nlm. nih.gov/nuccore/ MH645921 | NCBI GenBank, MH645921 |
| Schenk S, Bannister S, Sedlazeck FJ, Anrather D | 2019 | Combined transcriptome and proteome profiling reveals specific molecular brain signatures for sex, maturation and circalunar clock phase | https://www.ncbi.nlm. nih.gov/nuccore/ MH645922 | NCBI GenBank, MH645922 |
| Schenk S, Bannister S, Sedlazeck FJ, Anrather D | 2019 | Combined transcriptome and proteome profiling reveals specific molecular brain signatures for sex, maturation and circalunar clock phase | https://www.ncbi.nlm. nih.gov/nuccore/ MH645923 | NCBI GenBank, MH645923 |
| Schenk S, Bannister S, Sedlazeck FJ, Anrather D | 2019 | Combined transcriptome and proteome profiling reveals specific molecular brain signatures for sex, maturation and circalunar clock phase | https://www.ncbi.nlm. nih.gov/nuccore/ MH645924 | NCBI GenBank, MH645924 |
| Schenk S, Bannister S, Sedlazeck FJ, Anrather D | 2019 | Combined transcriptome and proteome profiling reveals specific molecular brain signatures for sex, maturation and circalunar clock phase | https://www.ncbi.nlm. nih.gov/nuccore/ MH678619 | NCBI GenBank, MH678619 |

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
