## [Decision Letter]

Thank you for submitting your article "Combined transcriptome and proteome profiling reveals specific brain states for sex, maturation and the circalunar phase" for consideration by *eLife*.

The Reviewing Editor evaluated the manuscript and the review that was received, and agreed with the positive overall inclination of the reviewer. This paper unquestionably represents a very important resource for a very intriguing biological phenomenon and *eLife* is, in principle, happy to provide a platform for publishing this as a Tools and Resources article.

However, the Reviewing Editor also agrees with the reviewer's request for a number of revisions. All three major comments are essential to address, but as you can see they will only require (very important) editorial changes and no further experimentation. In regard to major point 3, we urge you to discuss this issue with proper depth in the manuscript.

Overall assessment:

The manuscript "Combined transcriptome and proteome profiling reveals specific molecular brain signatures for sex, maturation and circalunar clock phase" by Schenk and coauthors provides several rich -omics datasets to define the molecular differences in the heads of the marine annelid *Platynereis dumerilii* between sexes, during maturation, and during different moon phases. The authors establish a novel method to extract both mRNAs and proteins from head/brain regions from the same sample enabling comparison between proteomic and transcriptomic results from the very same tissue. The authors generate a de novo head transcriptome to estimate transcript levels in the head under various conditions, perform differential gene expression analyses as well as subsequent downstream analyses for enrichments for these processes. >8000 differentially expressed transcripts (DETs) were identified during the maturation process, 640 DETs with a sex bias, and 63 DETs specific for one of two lunar phases. DET results were validated by qRT-PCR and in situ hybridization for a small number of selected genes with different signatures in regard to the three processes. Furthermore, 693 differentially expressed proteins (DEPs) were identified during maturation, 17 DEPS with a sex bias, and 261 DEPS during circalunar phases. Apparently the proteomic approach implicated more genes with significant changes during the two moon phases than the transcriptomic approach. The authors continued to investigate five circalunar DEPs genes that exhibited also correlated transcriptional changes within the RNA seq data. In situ hybridizations validated specific enrichments of the transcripts within distinct regions of the brain.

The generated datasets and analyses are of high quality and importance, and are a very significant advance to unravel molecular mechanisms that control the physiological changes in the brain during the circalunar phases. Furthermore the study provides a rich source to dissect and hone in on mechanisms during all three processes in more detail in the future. This is especially important in an organism like *Platynereis* in which circalunar biology can be functionally investigated under strict laboratory conditions, and dissected with an ever-increasing set of molecular tools thus promising a mechanistic understanding of circalunar regulation in the future.

Essential revisions:

1) Data accessibility: Although the authors provide a plethora of source data, it appeared difficult or impossible to find specific data especially the data that constitute Figure 3A and 5A. What are the 63 DETs for circalunar phase (Figure 3A)? I would like to see lists that rank the 8411 DET genes for maturation, the 640 DETs for sex bias, and the 63 DETs for circalunar including annotation and the mean for expression of relevant experiments e.g. for circalunar the mean expression for each of the two moon phases. This would be important to assess the data but also for the community to access this source.

Similarly what are the 17 DEPs for sex bias (Figure 5A)? I would like to see lists that rank the 510 DEPs for maturation, the 17 DEPs for sex bias, and the 261 DEPs for circalunar incl. annotation and the mean for expression between relevant conditions. Additional File 43 is an important file to document the authors' selection process, but a simplified version that simply lists the 261 DEPs including some annotation and selected relevant data would make these results and resource more easily accessible and usable.

As a resource the authors need to make these results easier to find and use.

2) Results: subsection “A maturation-stage and sex-locked lunar head transcriptome”, last paragraph: Spell out the logic of how the RNA-seq data of the different samples were used to generate the maturation, sex bias, and circalunar DETs. It is unclear in the main manuscript as written how the samples were used to generate the DETs shown in Figure 3A. Maybe add a scheme as Figure 2B that shows the general strategy how and which samples contributed to determining the number of DETs for each category maturation, sex, and circalunar. There is some explanatory simplified link missing here despite the details in the Materials and methods section.

3) It seems against common sense that this strategy unraveled over 261 DEPs for circalunar phases but only 17 DEPs for sex bias. This raises intuitively some red flags that I don't know how to resolve. The authors show that the proteomic approach identifies only the ~1000 most abundant proteins. What are the general transcript levels for the 261 DEPs? Are there within the top 1000 most frequent transcripts? Are there drastic exceptions? My expectation would be that all DEPs are in the ~ 2000 highest expressed transcripts within the RNA-seq data. Could DE analysis of proteomics data create 'artificially' high numbers of DEPs? Or does transcriptomic DE analysis throw out transcripts that are more variable between samples but nevertheless highly expressed? Variable high expression levels in one condition could make it less statistically significant, but the generally high transcript level could generate nevertheless a high synthesis rate of the corresponding protein. Thus, certain genes do not show up as DETs but as DEPs? It would be important to look into this a bit more and discuss it. I am uncertain whether a direct comparison of DETs and DEPs is valid or useful at this point, and therefore the last sentence at the end of the conclusion might be too strong. "Thus, the mechanisms responsible for circalunar reproductive timing likely operate and are coordinated more strongly at the post transcriptional, translational/post-translational level(s), than on transcript."

Related to this: Are the DETs biased towards generally lower or medium expression level genes? Would it be possible to 'mimic' the proteomic cut off in the transcriptome analysis? Does the differential gene expression analysis yield different results if restricted to the 1000 or 2500 or 5000 highest expressed genes?

---

## [Author Response]

Essential revisions:1) Data accessibility: Although the authors provide a plethora of source data, it appeared difficult or impossible to find specific data especially the data that constitute Figure 3A and 5A. What are the 63 DETs for circalunar phase (Figure 3A)? I would like to see lists that rank the 8411 DET genes for maturation, the 640 DETs for sex bias, and the 63 DETs for circalunar including annotation and the mean for expression of relevant experiments e.g. for circalunar the mean expression for each of the two moon phases. This would be important to assess the data but also for the community to access this source.Similarly what are the 17 DEPs for sex bias (Figure 5A)? I would like to see lists that rank the 510 DEPs for maturation, the 17 DEPs for sex bias, and the 261 DEPs for circalunar incl. annotation and the mean for expression between relevant conditions. Additional File 43 is an important file to document the authors' selection process, but a simplified version that simply lists the 261 DEPs including some annotation and selected relevant data would make these results and resource more easily accessible and usable.As a resource the authors need to make these results easier to find and use.

We apologize if these data were not as conveniently accessible as the reviewer preferred. The mentioned data were already present in the overall transcriptome and proteome files, and we had not anticipated that separate files would be needed. But we understand the point of the reviewer that having separate files helps to quickly access the specific transcripts and proteins. We now provide the requested lists as Figure 3—source data 12-14 and Figure 5—source data 19-21 (Additional File 43 is now Figure 5—source data 18). (Please note that the annotation for the transcriptome was generated using the blastx algorithm with an e-value cutoff of 10^-4^. There are many sequences that do not have similarity annotations. This is likely due to a combination of reasons: there is still no good proteome dataset for a closely related reference species in the public repositories, and the transcriptome assembly – even though superior to currently published datasets – likely contains still partial sequences.

2) Results: subsection “A maturation-stage and sex-locked lunar head transcriptome”, last paragraph: Spell out the logic of how the RNA-seq data of the different samples were used to generate the maturation, sex bias, and circalunar DETs. It is unclear in the main manuscript as written how the samples were used to generate the DETs shown in Figure 3A. Maybe add a scheme as Figure 2B that shows the general strategy how and which samples contributed to determining the number of DETs for each category maturation, sex, and circalunar. There is some explanatory simplified link missing here despite the details in the Materials and methods section.

We apologize that the description of the procedure was apparently not clear enough. There had been already a section under “Read mapping and differential expression analysis of RNA-Seq data and selection of top regulated candidates for validation”, which we now improved. We also included a detailed explanatory scheme as Figure 1—figure supplement 1 (and more clearly refer to Figure 1—source data 2).

3) It seems against common sense that this strategy unraveled over 261 DEPs for circalunar phases but only 17 DEPs for sex bias. This raises intuitively some red flags that I don't know how to resolve. The authors show that the proteomic approach identifies only the ~1000 most abundant proteins. What are the general transcript levels for the 261 DEPs? Are there within the top 1000 most frequent transcripts? Are there drastic exceptions? My expectation would be that all DEPs are in the ~ 2000 highest expressed transcripts within the RNA-seq data.

We understand this might appear counterintuitive, and – like the reviewer – we would have initially assumed that the detected proteins should belong to the most abundant transcripts. However, studies across various organisms and in different experimental contexts show that the correlation between the genes that are detected in the transcriptome and the corresponding proteome is relatively low. As a matter of fact, our correlation coefficient (R=0.48) is at the upper end of what has been published for such analyses in organisms. To provide concrete examples:

Casas-Vila et al., 2017, found only a moderate correlation (maximum R=0.5) for the developmental transcriptome/proteome in the fruitfly *Drosophila melanogaster.* Comparison of transcriptome vs. proteome for sexual differences in the nematode *C.elegans* by Tops et al., 2010, results in an R=0.41. Likewise, Grün, D et al., 2014, compare protein and mRNA expression levels for multiple stages of *C.elegans* development. While they do not provide absolute numbers across all stages (for direct comparison to our numbers), their correlations between mRNA and protein levels across different developmental stages (see their Figure 4C, p-570) are much lower than ours. These authors attribute the effect (at least in part) to post-transcriptional regulation. Finally, also in line with all these observations, Schrimpf, SP et al. (PLoS Biol. 2009 Mar 3;7(3):e48.) already noticed that the abundance of orthologous proteins across species is better correlated than the protein vs. transcript abundances within the same species. We had mentioned this aspect already briefly in the previous version of the manuscript, but now extended it, including the above-mentioned citations (see “To further compare to existing datasets…”).

All the cited studies provide evidence that the assumption that the detectable DEP should be among the highest approximately 2000 expressed RNA transcripts is incorrect. Indeed, consistent with these studies, we find that DEPs for maturation, sex and lunar distribute as a cloud across the DETs corresponding to all detected proteins. We do not detect a particular bias. To make this point clearer, we now included these analyses in the initial transcriptome-proteome correlation figure (Figure 5—figure supplement 1).

Could DE analysis of proteomics data create 'artificially' high numbers of DEPs? Or does transcriptomic DE analysis throw out transcripts that are more variable between samples but nevertheless highly expressed? Variable high expression levels in one condition could make it less statistically significant, but the generally high transcript level could generate nevertheless a high synthesis rate of the corresponding protein. Thus, certain genes do not show up as DETs but as DEPs? It would be important to look into this a bit more and discuss it. I am uncertain whether a direct comparison of DETs and DEPs is valid or useful at this point, and therefore the last sentence at the end of the conclusion might be too strong. "Thus, the mechanisms responsible for circalunar reproductive timing likely operate and are coordinated more strongly at the post transcriptional, translational/post-translational level(s), than on transcript."

We have no evidence that particularly the lunar (and also maturation) transcript vs. proteins have a higher variability on transcript level than on proteome level. But as this might escape our detection to some extent (e.g. how variable would the transcriptome vs. proteome have to be in order to get such an effect?), we now mention this as a point that should be kept in mind in the revised Discussion section (“Alternatively, it is also possible that the level of variability is higher in the transcriptome than in the proteome, which…”).

It is clear that further quantitative transcriptomic and proteomic studies will be beneficial to better understand the circalunar effects on the worms. However, we think that our data already provide significant conceptual advance with respect to the possible functional mechanism and impact of the circalunar clock:

Combined transcriptome and proteome comparisons have previously been taken as a strong indication for post-transcriptional regulation, see e.g. Grün et al., 2014, “For most genes, protein abundances could not be explained by changes of the transcript level, suggesting ubiquitous posttranscriptional regulation….” Also see Zappulo A et al. (Nat Commun. 2017 Sep 19;8(1):583) for the usage of such data to conclude on spatially specific post-transcriptional regulation. But to take the reviewer’s comments into account, we have now softened the respective statement as follows: “Thus, the mechanisms responsible for circalunar reproductive timing *possibly* operate and *might be* coordinated more strongly at the post transcriptional….”

We also include a clearer statement that our study can only be seen as a first glimpse into these comparisons and that further analyses across more timepoints (at all levels) will be required to get a better understanding.

Related to this: Are the DETs biased towards generally lower or medium expression level genes? Would it be possible to 'mimic' the proteomic cut off in the transcriptome analysis? Does the differential gene expression analysis yield different results if restricted to the 1000 or 2500 or 5000 highest expressed genes?

We thank the reviewer for these suggestions and performed the requested analyses. The results are detailed below, and we also included them in the Results part of the manuscript (Results subsection “Additional proteomic analyses confirm the large change in protein abundance between different circalunar phases”, from “Finally, we tested if the high numbers of circalunar regulated proteins could be explained by the lower number of proteins compared to the high transcriptome number,…” and subsection “Mass spectrometry” for methodological description).

In summary, focusing the analyses on the top 5000 or 1000 expressed transcripts can explain an about 3-4.3-fold increase in significantly regulated DET. This is likely due to a “relaxation” of the multiple testing correction due to lower total numbers to correct for. Interestingly, this 3-4.3-fold increase in enrichment with lower numbers of tested transcripts covers the increase we observe for protein regulation across maturation. In contrast, it does not explain the about 245-fold increase we observe between circalunar transcriptome and proteome.

Circalunar:

5000 most highly expressed transcripts:

20 DET = 0.4% (compared to 0.12% in the complete transcriptome set) – 3.3-fold enrichment

1000 most highly expressed transcripts:

4 DET = 0.4% (compared to 0.12% in the complete transcriptome set) – 3.3-fold enrichment

Sex differences:

5000 most highly expressed transcripts:

262 DET = 5.2% (compared to 1.22% in the complete transcriptome set) – 4.3-fold enrichment

1000 most highly expressed transcripts:

52 DET = 5.2% (compared to 1.22% in the complete transcriptome set) – 4.3-fold enrichment

Maturation:

5000 most highly expressed transcripts:

2435 DET = 48.7% (compared to 16.5% in the complete transcriptome set) – 3.3-fold enrichment

1000 most highly expressed transcripts:

546 DET = 54.6% (compared to 16.5% in the complete transcriptome set) – 3.3-fold enrichment

As we did these calculations, we noticed that the programme versions of DESeq and EdgeR have meanwhile been updated. We thus also re-verified the stated changes for the total transcriptome with the updated versions. The newer versions are apparently slightly more stringent, but show only very minor impacts on the total numbers.

Lunar regulation: 0.10% vs 0.12%

Sex specific regulation: 1.22% vs. 1.23%

Maturation regulated: 16.17% vs. 16.52%